# Ubiquitin-Specific Proteases (USPs) and Metabolic Disorders

**DOI:** 10.3390/ijms24043219

**Published:** 2023-02-06

**Authors:** Hiroshi Kitamura

**Affiliations:** Laboratory of Comparative Medicine, School of Veterinary Medicine, Rakuno Gakuen University, Ebetsu 069-8501, Japan; ktmr@rakuno.ac.jp; Tel.: +81-11-388-4781

**Keywords:** ubiquitin-specific protease, metabolic disorder, obesity, diabetes, insulin resistance, non-alcoholic fatty liver disease, atherosclerosis, cardiovascular disease

## Abstract

Ubiquitination and deubiquitination are reversible processes that modify the characteristics of target proteins, including stability, intracellular localization, and enzymatic activity. Ubiquitin-specific proteases (USPs) constitute the largest deubiquitinating enzyme family. To date, accumulating evidence indicates that several USPs positively and negatively affect metabolic diseases. USP22 in pancreatic β-cells, USP2 in adipose tissue macrophages, USP9X, 20, and 33 in myocytes, USP4, 7, 10, and 18 in hepatocytes, and USP2 in hypothalamus improve hyperglycemia, whereas USP19 in adipocytes, USP21 in myocytes, and USP2, 14, and 20 in hepatocytes promote hyperglycemia. In contrast, USP1, 5, 9X, 14, 15, 22, 36, and 48 modulate the progression of diabetic nephropathy, neuropathy, and/or retinopathy. USP4, 10, and 18 in hepatocytes ameliorates non-alcoholic fatty liver disease (NAFLD), while hepatic USP2, 11, 14, 19, and 20 exacerbate it. The roles of USP7 and 22 in hepatic disorders are controversial. USP9X, 14, 17, and 20 in vascular cells are postulated to be determinants of atherosclerosis. Moreover, mutations in the *Usp8* and *Usp48* loci in pituitary tumors cause Cushing syndrome. This review summarizes the current knowledge about the modulatory roles of USPs in energy metabolic disorders.

## 1. Introduction

Age, obesity, and inactivity increase blood pressure, blood glucose, blood triglycerides and cholesterol, and body fat levels. Although the definition of metabolic syndrome differs between groups, the excessive accumulation of lipids in visceral adipose tissue combined with hypertension, hyperglycemia, or hyperlipidemia can be defined as metabolic syndrome [1,2,3]. Patients with metabolic syndrome exhibit a high prevalence of cardiovascular disease and stroke [4,5], both of which are major causes of death [5,6,7]. Moreover, metabolic syndrome increases insulin resistance, leading to type 2 diabetes mellitus (T2DM). T2DM worsens metabolic syndrome due to further increases in blood glucose and lipids [8,9]. While the number of people with metabolic disorders, including metabolic syndrome, is increasing due to the COVID-19 pandemic, such an increase was already previously observed in the Western world and has become a worldwide issue due to the spread of Western diets [10]. Metabolic syndrome currently affects over a billion people globally [10]. Metabolic disorders primarily occur due to dysfunction of energy metabolism–competent tissues, such as skeletal muscle, liver, adipose tissues, pancreatic islets, and the hypothalamus [11,12,13], and subsequent damage to the vascular system and nervous system is observed [11,14]. Local chronic inflammation and/or excessive oxidative stress are proposed to be responsible for the incidence of metabolic disorders [11,15,16,17,18]. To date, numerous molecules, including growth factors, cytokines, antioxidants, chaperones, protein kinases, transcriptional regulatory proteins, and histone modifiers, have been identified as key molecules that determine the progression of metabolic disorders [19,20,21,22,23,24,25]. Specifically, accumulating evidence indicates that enzymes catalyzing post-translational regulation, such as phosphorylation [26], oxidation [27], O-GlcNacylation (addition of O-linked N-acetylglucosamine chains to proteins) [28], and ubiquitination [29], have pivotal roles in the pathogenesis of metabolic disorders.

Protein ubiquitination is one of many major post-translational modifications of proteins, and constitutes a reversible reaction regulated by ubiquitin ligases and deubiquitinating enzymes (DUBs). DUBs are categorized based on the structure of their catalytic domain and consist of the following eight subfamilies: ubiquitin-specific proteases (USPs), ubiquitin C-terminal hydrolases (UCHs), ovarian tumor domain-containing proteases (OTUs), Machado-Joseph disease protein domain protease (MJDs), JAB1/MPN/Mov34 metalloproteases (JAMMs), monocyte chemotactic protein-induced protein (MCPIPs), motif interacting with ubiquitin-containing novel DUB (MINDYs), and zinc finger and UFSP domain protein (ZUFSP) [30]. USPs constitute the largest DUB subfamily, comprising 58 members of vertebrates [31]. USPs are cysteine proteases that have three parts of the conserved USP domain, termed the fingers, thumb, and palm [30,32]. The size of the USP domain ranges from less than 300 to more than 800 amino acids [30]. Accessory domains, including the ubiquitin-associated domain (UBA), the ubiquitin-interacting motif (UIM), and the zinc finger ubiquitin-specific protease domain (ZnF-UBP), as well as terminal extensions, have been proposed to confer substrate specificity to USPs [30,33].

Because several USPs control cell cycle progression, DNA damage repair, and cancer-related cellular signaling, their roles in oncogenesis have been investigated [30,34,35,36], and knowledge about their pathological and physiological roles has recently begun accumulating [37]. In this review, we sought to summarize the roles of USPs in metabolic disorders, such as obesity, diabetes mellitus, non-alcoholic fatty liver disease (NAFLD), and atherosclerosis. Additionally, we discuss somatic mutations in USPs in the context of Cushing syndrome, which is characterized by the malfunction of energy metabolism.

## 2. Obesity and Adipogenesis

Excessive intake of carbohydrates or lipids results in the storage of triglycerides in white adipose tissue. White adipose tissue adapts to the presence of excess nutrients by increasing cellular volume (hypertrophy) and increasing the amount of mature adipocytes by evoking differentiation from preadipocytes (hyperplasia) [38]. Importantly, hypertrophic adipocytes tend to undergo necrotic death, which leads to fibrosis, chronic inflammation, and secretion of harmful adipokines [38]. Dysfunction of white adipose tissue causes pathological increases in lipids and glucose in circulating blood, resulting in metabolic disorders such as diabetes mellitus, NAFLD, cardiovascular diseases, and some types of cancers [39]. To date, several USPs have been shown to modify obesity via the control of adipocyte differentiation, i.e., adipogenesis.

The transcriptional regulatory factors that are responsible for adipogenesis have been well documented. Following stimulation with adipogenic hormones, such as insulin, glucocorticoids, and 3,3,5-triiodothyronine, CCAAT-enhancer-binding protein (C/EBP)β initiates the differentiation of preadipocytes [40]. In turn, C/EBPα and peroxisome proliferator-activated receptor γ (PPARγ) cooperatively induce genes required for the function of mature adipocytes [40]. C/EBP homologous protein (CHOP) inhibits the effects of C/EBPs, and thereby inhibits adipogenesis [41]. Hunag et al. showed that the COP9 signalosome (CSN)-cullin RING ubiquitin ligase (CRL) super-complex destabilizes the CHOP1 protein, which facilitates adipogenesis [42]. The authors also suggested that USP15 promotes adipogenesis because the overexpression of USP15 protects a CRL component, Keap1, from autoubiquitination [42,43].

In addition to C/EBPs and PPARγ, activating transcription factor 4 (ATF4) is also believed to be essential for adipogenesis [44]. ATF4 elevates sterol regulatory element binding protein 1c (SREBP1c), which is a known vital transcription factor for lipogenesis [45]. ATF4-induced SREBP1c was shown to be regulated at the post-translational level, since *Atf4* siRNA increased *Srebp1c* mRNA [45]. *Atf4* siRNA significantly downregulated the expression of *Usp4, 7, 8, 11, 12, 19, 20, 21, 22, 27, 28, 30 31, 32, 36, 42, 44, 46, 50, 53,* and *54*; however, the expression of *Usp7*, also called herpesvirus-associated ubiquitin-specific protease (HAUSP), was the most significantly repressed [45]. Conversely, overexpression of *Atf4* markedly upregulated *Usp7* mRNA [45]. Moreover, overexpression of *Usp7* recovered the impairment of adipogenesis by *Atf4*-deficiency, in agreement with SREBP1c accumulation [45]. Overall, USP7 is involved in ATF4-induced SREBP1c accumulation and subsequent adipocyte differentiation.

Administration of 3′3-diindolylmethane (DIM, 10 or 50 mg/kg), a chemical abundantly contained in cruciferous vegetables, significantly attenuated body weight gain in mice on high-fat diets [46]. Accordingly, DIM administration also reduced epididymal adipose tissue weight and halted the differentiation of cultured adipocytes at an early stage [46]. In addition, DIM-treated cells had reduced cyclin D protein, which is necessary for early-stage adipogenesis [46]. The reduction of cyclin D might be caused by the inhibition of USP2, which catalyzes the deubiquitination of cyclin D [47]. Furthermore, since USP2 stabilizes fatty acid synthase (FASN) in cancerous cells [48], USP2 may facilitate lipogenesis via FASN induction in other cell types, including adipocytes.

*Usp19* knockout (KO) mice have remarkably decreased fat pads by 24 months of age [49]. In *Usp19*KO mice, preadipocytes failed to differentiate, and the induction of adipogenic genes such as PPARγ, C/EBPα, fatty acid binding protein 4 (FABP4), adiponectin, and leptin were impaired [49]. Although food intake was comparable between *Usp19*KO and wild-type control mice, oxygen consumption and energy expenditure were significantly increased by *Usp19* deficiency. Thus, adipocyte USP19 is involved in obesity.

Song et al. identified USP20 as a stabilizer of HMG-CoA reductase (HMGCR), which is a rate-limiting enzyme for cholesterol biosynthesis [50]. They showed that USP20 is critical for the feeding-induced increase in HMGCR in the liver [50]. After insulin stimulation, USP20 is phosphorylated by the mammalian target of rapamycin complex 1 (mTORC1) and then stabilizes the HMGCR complex on the endoplasmic reticulum via interaction with Gp78 [50]. On the other hand, *Hmgcr* deficiency–induced accumulation of HMG-CoA accelerated energy consumption [50], because succinate, which is produced from HMG-CoA, activates thermogenesis [51,52]. Accordingly, hepatic *Usp20* deficiency decreased fat mass as well as the adipocyte area in adipose tissue [50]. This agrees with findings that a USP20 inhibitor called GSK2643943A enhanced body weight gain and oxygen consumption in mice [50]. These results collectively indicate that USP20 is a therapeutic target of adiposity.

It has been suggested that USP53, in contrast to USP2, 7, 15, 19, and 20, may have beneficial effects on obesity. Bolton et al. investigated potential biomarkers of successful weight control during a dietary intervention consisting of a low-calorie diet for eight weeks, followed by a cafeteria diet for six months [53]. They identified a total of 209 genes whose expression levels differed between baseline and the end of the intervention. Individuals with high USP53 expression in adipose tissue had a markedly decreased body mass index during the intervention. Additionally, analysis of the expression of quantitative trait loci indicated that genetic polymorphisms of *USP53* are associated with the transcript level in adipose tissue. Thus, USP53 is likely to be a determinant of whether a given person’s constitution is sensitive to calorie-restriction interventions, although the molecular mechanisms underlying USP53-mediated regulation remain elusive.

## 3. Diabetes Mellitus and Insulin Resistance

Diabetes mellitus is a disorder of carbohydrate metabolism caused by the hypo-production of insulin in pancreatic β-cells and/or insufficient insulin action in carbohydrate metabolism–competent organs such as the liver, skeletal muscle, and white adipose tissue [54]. Diabetes is classically categorized into type 1 diabetes mellitus (T1DM) and T2DM.

### 3.1. T1DM

T1DM is characterized by the destruction of pancreatic β-cells primarily by autoimmunity [55]. T1DM patients exhibit extremely low levels of circulating insulin but are sensitive to administrated insulin. From the viewpoint of T1DM as an autoimmune disease, the findings from some studies indicate the involvement of immune-associated USPs in T1DM pathology. Because CRISPR/Cas9-dependent KO of the *Usp22* gene decreased the occurrence of forkhead box P3 (FoxP3)^+^ cells, it appears that USP22, a deubiquitinating enzyme in the SAGA chromatin modifying complex, stabilizes FoxP3 [56]. FoxP3 is a pivotal transcription factor for the development of regulatory T cells (Tregs) and is primarily a suppressor of autoimmunity [57]. Since USP22 is required for Treg induction [56], it might have an immunosuppressive role in T1DM through the induction of Tregs.

Type I interferons (IFNs) in the pancreas, such as IFN-α and IFN-β, increase the risk of T1DM, and type I IFNs appear to be a trigger of T1DM. Santin et al. reported that siRNA against USP18 increased type I IFN–elicited events in β-cells, such as signal transducer and activator of transcription (STAT) 1/2–induced inflammatory responses, and mitochondrial pathway-driven apoptosis [58]. Moreover, *Usp18* knockdown also strengthened chemokine induction via the increase of melanoma differentiation-associated protein 5 (MDA5), which is a sensor protein of viral double-stranded RNA [58]. Given that viral infection itself stimulates T1DM, USP18 might be a critical suppressor of T1DM.

### 3.2. T2DM

Elevation of circulating fasting glucose levels, triglycerides, saturated fatty acids, inflammatory cytokines, and advanced glycation end products (AGEs), causes β-cell dysfunction and insulin resistance [9,18,21], both of which are major causes of T2DM [54,59,60]. T2DM is highly prevalent worldwide. According to a previous epidemiological report based on data from the Institute of Health Metrics at Seattle, T2DM affected 462 million individuals in 2017 (~6.28% of the then global population) [61]. This report also predicted that the global prevalence of T2DM might increase to 7% by 2030 [61]. Other reports documented a global diabetic prevalence of 10.5% in people aged 20–79 years, and projected an increase to 12.2% by 2045 [62]. Considering that more than 90% of the diabetic burden is caused by T2DM [63], approximately 9% (~482 million) of individuals are predicted to be affected by T2DM by 2045.

After the binding of insulin to the insulin receptor, the tyrosine kinase domain of the insulin receptor β-chain phosphorylates insulin receptor substrate 1 (IRS1), leading to the phosphorylation of phosphatidyl inositol 3 kinase (PI3K) [26]. Subsequently, PI3K catalyzes the conversion of phosphatidylinositol-3,4-diphosphate to phosphatidylinositol (3,4,5)-triphosphate, which promotes the phosphorylation of 3-phosphoinositide-dependent protein kinase 1 (PDK1) [26]. Then, PDK1 phosphorylates the 308th amino acid (threonine) of Akt [26]. Akt provokes a variety of insulin-dependent cellular responses, including the translocation of glucose transporter (GLUT) 4 onto the cell surface [64]. In the case of insulin resistance, insulin signal transduction is disrupted by protein modifications such as phosphorylation and dephosphorylation [26], O-GlcNAcylation [65], and ubiquitination [33] of either signaling molecule. Some USPs have been proposed to modulate insulin signaling in several tissues through their deubiquitination activity (Figure 1).

#### 3.2.1. Pancreatic β-Cells

Since β-cells express GLUT2 abundantly, hyperglycemia promotes glucose uptake and subsequent glycation and mitochondrial oxidative phosphorylation [66]. As a result, reactive oxygen species (ROS) accumulate in β-cells as a byproduct [66]. Because antioxidants are scarce in β-cells, oxidative stress easily occurs in these cells [67], and β-cell dysfunction is evident in patients with severe T2DM. Chronic exposure to high concentrations of glucose stimulates the acetylation of the mitochondria-specific hydrogen peroxide scavenger peroxiedoxin-3 (PRDX3) in β-cell–like INS-1^+^ cells, due to the degradation of sirtuin (Sirt)-1, a deacetylase [68]. Hyperglycemia has been shown to decrease intracellular USP22, which stabilizes Sirt-1 through direct interactions [68]. This report also indicated that p38 mitogen-activated kinase (MAPK) is involved in decreasing USP22 expression under hyperglycemic conditions [68]. In support of this observation, teneligliptin, an inhibitor of dipeptidyl peptidase-4, overcame the p38-dependent reduction in USP22 expression [68]. As a consequence, the accumulation of Sirt-1 conferred enhanced redox activity in β-cells through the deacetylation of PRDX3 [68]. Therefore, USP22 plays a protective role in β-cell dysfunction of T2DM patients.

High concentrations of glucose, palmitate, and proinflammatory cytokines (mixtures of interleukin (IL)-1β and IFNγ or IL-1β, IFNγ and tumor necrosis factor-α (TNF-α)) induce β-cell apoptosis [69]. The high glucose, high glucose and palmitate, or cytokine-induced apoptosis of β-cells was significantly suppressed by short hairpin RNA (shRNA) specific for *USP1*, or USP1 chemical inhibitors ML323 or SJB2-043, as characterized by the repression of caspase-3 cleavage and poly ADP-ribose polymerase [69]. The proapoptotic effects of USP1 are attributed to DNA damage, since the inhibition of USP1 decreased γ-H2AX-positive β-cells even after treatment with apoptosis inducers [69]. Coincidentally, treatment of isolated human islets from T2DM patients with USP1 blockers repressed the DNA damage response, along with a decrease in apoptosis [69]. These results indicate that USP1 is involved in the DNA damage-induced apoptosis of islet β-cells in T2DM patients.

#### 3.2.2. Adipose Tissue

As noted above, USP19 is required for adipogenesis [49]. Thus, ablation of *Usp19* in mice substantially decreased fat mass, and led to a reduction of circulating fatty acids, including palmitate [49]. Since palmitate is well known to be an inducer of insulin resistance [70], USP19 can be considered to promote insulin resistance via adipogenesis. Accordingly, *Usp19*KO mice showed a significant decrease in fasting insulin levels, along with the restoration of glucose and insulin sensitivities [49]. Moreover, *Usp19* deficiency reduces insulin resistance in the liver and skeletal muscle [49]. Thus, adipocyte USP19 exacerbates T2DM.

Chronic inflammation of visceral adipose tissue is believed to be a trigger of T2DM [71]. Correspondingly, lipid-laden hypertrophic adipocytes secrete “bad” adipokines, miRNAs, and metabolites, all of which exacerbate insulin resistance [72,73]. Inflammatory macrophages accumulate around necrotic adipocytes in a crown-like structure and produce inflammatory cytokines, such as TNF-α and IL-6, which also exacerbate insulin resistance [74,75,76]. Kitamura et al. reported that *USP2* knockdown in human macrophage-like HL-60 cells displayed an increased expression of proinflammatory cytokines in response to inflammatory stimuli, while macrophages isolated from *Usp2* transgenic mice had repressed cytokine expression [77]. *Usp2* knockdown also selectively increased the expression of FABP4, plasminogen activator inhibitor 1, and several chemokines, all of which are associated with the progression of T2DM [78]. This agrees with the finding that adipose tissue macrophages in obese individuals have significantly decreased levels of USP2 [78]. Conversely, macrophage-selective *Usp2* transgenic mice rectified high fat diet–induced adipose tissue inflammation, resulting in the retention of insulin sensitivity, especially in the skeletal muscle and liver [79].

#### 3.2.3. Skeletal Muscle

Skeletal muscle is known to be the organ with the highest glucose consumption, accounting for ~75% of glucose uptake from the circulation [80]. Muscle fibers can be roughly classified into fast, slow, and fast intermediate fibers [81]. Fast fibers, such as myosin heavy chain (Myh) IIB and IIX-positive fibers, preferably obtain ATP by glycolysis, while slow fibers, such as Myh I and IIA-positive fibers produce ATP by oxidative phosphorylation in mitochondria [82]. Since slow fibers consume more carbohydrates than fast fibers, conversion of fast to slow fibers prevents obesity and subsequent T2DM onset [83]

A recent paper demonstrated that ablation of the *Usp21* gene increased mitochondria levels in skeletal muscle through the induction of peroxisome proliferator-activated receptor γ coactivator (PGC) 1α, causing the proportion of oxidative fibers characterized by Myh I and IIa in skeletal muscle to become predominant [84]. *Usp21* deficiency–induced alterations in muscle fibers were reversed by overexpression of *Usp21*. These results indicate that muscular USP21 seems to aggravate carbohydrate metabolism by interrupting mitochondrial activation. In support of this interpretation, the upregulation of USP21 expression in skeletal muscle was observed in obese individuals [84]. Moreover, fasting glucose levels were significantly correlated with muscular *Usp21* expression in high fat diet–fed mice [84]. Conversely, skeletal muscle-specific ablation of *Usp21* improves obesity and insulin sensitivity [84]. Mechanistically, this report identified DNA-dependent protein kinase catalytic subunits (DNA-PKcs) and ATP citrate synthase (ACLY) as substrates of USP21 [84]. Overexpression of *Usp21* interfered with proteasome-dependent digestion of DNA-PKcs and ACLY via removal of the polyubiquitin chain [84]. Subsequently, accumulated DNA-PK and ACLY blunted the activation of AMP-activated kinase (AMPK), which is responsible for mitochondrial fuel activation [84].

It has been etiologically and experimentally shown that calorie restriction improves age-related disease prognoses, including metabolic disorders [85,86]. The beneficial effects of calorie restriction account for the increase of AMPKα2 subunit expression in skeletal muscle [87]. Accordingly, *Ampka2*KO mice have abrogated calorie restriction–stimulated glucose uptake in skeletal muscle, even after insulin stimulation [87]. In the presence of serum from calorie-restricted individuals, the stability of the AMPKα2 subunit increased in cultured C2C12 myocytes due to its deubiquitination [87]. In this case, siRNA for *Usp9X*, but not for *Usp5* and *7*, blocked ubiquitination-mediated degradation of the AMPKα2 subunit [87]. Therefore, muscular USP9X is likely to be a key determinant of the anti-diabetic effect of calorie restriction.

To date, numerous humoral factors and metabolites have been shown to modify insulin signaling in skeletal muscle [88,89,90,91]. Adrenaline and noradrenaline bind the β_2_ adrenoreceptor and provoke cAMP accumulation through adenylate cyclase activation [92]. Consequently, cAMP-dependent kinase (PKA) stimulates the phosphorylation of Akt, resulting in the potentiation of insulin signaling [93]. A previous paper employing overexpression and knockdown models demonstrated that USP20 and 33 have the potential to remove polyubiquitin chains from the β_2_ adrenoreceptor [94]. Therefore, USP20 and 33 seem to sustain insulin signaling by stabilizing the β_2_ adrenoreceptor. Another recent paper reported that the secreted form of endoplasmic reticulum aminopeptidase 1 (ERAP1) released from hepatocytes lowered insulin sensitivity in skeletal muscle [95]. In this case, ERAP1 promoted the dissociation of USP33, but not USP20, from the β2 adrenoreceptor, resulting in polyubiquitination of the receptor. By disrupting USP33-dependent stabilization of the β2 adrenoreceptor, ERAP1 exacerbated insulin signaling in skeletal muscle, in parallel with defects in insulin sensitivity at an individual level [95].

#### 3.2.4. Liver

In addition to modulating carbohydrate metabolism in other organs by hepatokines [96], the liver also controls blood glucose levels by glycogen synthesis and glycogenolysis [97]. Additionally, the liver produces glucose via gluconeogenesis from non-carbohydrate precursors, such as pyruvate, amino acids, and lactate [97]. In diabetic patients, hepatic glucose synthesis is promoted by insulin resistance and augmented by catabolic hormones such as glucagon and glucocorticoids [98]. To date, several hepatic USPs have been shown to modulate glucose metabolism and change circulating glucose levels.

Since USP7 increases the stability of IRS1 by deubiquitination in hepatocytes, USP7 prolongs hepatic insulin signaling [99]. USP7 is preferably associated with membrane protein phosphate inorganic transporter 1 (PiT1) in hepatocytes. Thus, hepatocyte-specific KOs of *Slc20a1* (the gene encoding PiT1) result in the accumulation of free USP7 in the cytoplasm. Subsequently, USP7 promotes the deubiquitination of IRS1 by direct interaction, and thereby prolongs insulin signaling [99]. In agreement with these observations, hepatocyte-specific *Slc20a1*KO mice were protected from high fat diet–induced insulin resistance due to enhanced insulin signaling [99]. In addition to IRS1, ectopic expression of *Usp7* also stabilized PPARγ by direct deubiquitination [100]. Infection with *Usp7-*overexpressing adenovirus markedly elevated PPARγ levels in hepatocytes and increased *Usp7* expression [100]. Although *Usp7* overexpression augmented lipid accumulation in hepatocytes, it significantly decreased blood glucose, free fatty acids, and triglyceride levels [100]. In agreement with these observations, hepatic USP7 also potentiates insulin sensitivity [100]. Moreover, overexpression of *Usp7* decreased polyubiquitination of FoxO1, a transcription factor that regulates the transcription of rate-limiting enzymes for gluconeogenesis, including glucose 6-phosphatase catalytic subunit (G6PC) and phosphoenolpyruvate carboxykinase 1 (PCK1) [101]. USP7 also inhibits FoxO1-dependent transcription by removing monoubiquitin from FoxO1 [101]. Accordingly, hepatic *Usp7* overexpression strongly decreased the levels of nuclear FoxO1 and *G6pc* and *Pck1* in the livers of C57BL/6 mice. As a result, mice infected with the *Usp7-*expressing construct showed improved results in pyruvate tolerance tests [101]. Thus, USP7 prevents T2DM by (1) improving insulin sensitivity, and (2) inhibiting gluconeogenesis.

Previous reports have demonstrated that two additional USPs, namely USP2 and USP14, modulate hepatic gluconeogenesis [102,103]. Hepatic gluconeogenesis exhibits periodicity over the light/dark cycle [104,105]. In the suprachiasmatic nucleus, USP2 expression is controlled by the brain and muscle arnt-like 1 (BMAL-1)/Clock complex and positively regulates circadian glucose metabolism through gluconeogenic enzymes such as G6PC and PCK1 [102,106]. Correspondingly, infection with an adenovirus encoding *Usp2* shRNA significantly lowered blood glucose, blood insulin, hepatic glycogen content, and insulin sensitivity in high fat diet–fed mice, suggesting that USP2 plays a role in aggravating T2DM pathogenesis [102]. At the mechanistic level, overexpression of *Usp2* was shown to stabilize C/EBP-α, a transcription factor that induces the expression of 11β-hydroxysteroid dehydrogenase 1 (HSD1) [102]. Given that HSD1 converts circulating inactive cortisone to active cortisol [107], hepatic USP2 causes hyperglycemia by activating glucocorticoid signaling.

Sustained low-grade endoplasmic reticulum (ER) stress induced by low doses of tunicamycin (200 μg/kg) treatment for two weeks increased hepatic gluconeogenesis and resulted in hyperglycemia [103]. The tunicamycin-treated animals did not have exacerbated insulin sensitivity, suggesting that low-grade ER stress is unlikely to cause insulin resistance in hepatocytes [103]. On the other hand, sustained ER stress robustly increased ATF4-activated USP14 expression and accelerated hepatic gluconeogenesis [103]. In contrast, *Usp14* knockdown lowered blood glucose levels, induced glucose intolerance, and downregulated gluconeogenic genes [103]. Overexpression and knockdown of *Usp14* respectively increased and decreased 3′,5′-cyclic monophosphate-responsive element binding protein (CBP) in the livers of obese mice [103]. Given that CBP is a downstream component of the glucagon signaling pathway, these observations suggest that hepatic USP14 promotes gluconeogenesis by strengthening glucagon signaling.

Besides regulating gluconeogenesis, hepatic USP14 also decreases insulin sensitivity. High fat diet–fed obese mice and leptin receptor–deficient *db*/*db* mice displayed decreased expression of *Usp14* in the liver [108]. Combinatory proteomics identified FASN as an interactor of USP14 in the liver [108]. *db*/*db* mice that were infected with an adenovirus continuously expressing shRNA for *Usp14* had substantially decreased blood insulin and glucose levels [108]. Additionally, *Usp14* silencing ameliorated insulin resistance in these mice [108].

There have been reports indicating that other USPs directly modify insulin sensitivity in hepatocytes. USP4 was downregulated in the livers of high fat diet–fed mice and *ob*/*ob* mice [109]. High fat diet–induced insulin resistance was aggravated and reversed by hepatocyte-specific *Usp4* deficiency and *Usp4* overexpression, respectively, as assessed by fasting insulin levels, homeostasis model assessment-insulin resistance (HOMA-IR), glucose tolerance tests, and insulin tolerance tests [109]. USP4 directly deubiquitinated transforming growth factor-β activated kinase 1 (TAK1), resulting in TAK1 dephosphorylation [109]. As a result, the nuclear factor of κ light polypeptide gene enhancer in B-cells (NF-κB) and c-Jun N-terminal kinase (JNK) signaling cascades were inhibited [109]. NF-κB and JNK participate in inflammatory responses and/or cellular stress responses, both of which are well known to evoke insulin resistance [110]. Hence, USP4 is thought to attenuate insulin resistance by reducing inflammatory and cellular stress. USP18 is also downregulated in a mouse model of T2DM [111], and in transgenic mice, obesity-induced insulin resistance was ameliorated and aggravated by hepatocyte-selective *Usp18* overexpression and deficiency, respectively [111]. Assessments using adenoviral *Usp18* and hepatocyte-selective *Usp18*KO mice revealed that USP18 suppressed the TAK1-NF-κB/JNK signaling axis [111]. Therefore, hepatic USP4 and USP18 improve insulin sensitivity by suppressing inflammatory and cellular stress signals. In agreement with these data, exercise increased hepatic USP4 protein and dual-specificity phosphatase14 and decreased tripartite motif 8 expression, which are suppressors and promoters of TAK1 phosphorylation, respectively [112]. Given that exercise reverses high fat diet–inhibited phosphorylation of IRS1 and Akt in the liver, it improves insulin sensitivity by inducing USP4.

A report suggested that USP4 might also mediate the beneficial effects of traditional medicine in the context of insulin signaling. Gastrodin, a chemical derived from *Gastrodia elata* Blume (Orchidaceae), significantly decreased dexamethasone-induced ubiquitination of insulin receptors in HepG2 cells, and concomitant increases of USP4 mRNA and protein were also observed [113]. Mechanistically, gastrodin suppressed the association of GATA binding protein 1 (GATA1) with the *Usp4* promoter, suggesting that gastrodin likely increases *USP4* transcription in a GATA1-dependent manner [113]. Thus, gastrodin might avoid the proteasome-dependent digestion of insulin receptors in hepatocytes via USP4 induction.

USP10 is another USP that restores insulin sensitivity. As with *Usp4* and *18*, the expression of *Usp10* transcript gradually decreased in the livers of high fat diet–fed mice [114]. Knockout and overexpression of *Usp10* in the liver promoted or attenuated high fat diet–induced insulin resistance, respectively [114]. Similarly, *Usp10* overexpression dampened insulin resistance in *ob*/*ob* mice [114]. The beneficial effects of USP10 on insulin sensitivity were abolished by *Sirt6* deficiency, and overexpression of *Sirt6* strongly abrogated the insulin resistance observed in *Usp10*KO mice [114]. Finally, USP10 directly stabilized Sirt-6 in in vitro models [114]. Therefore, the anti-diabetic activity of hepatic USP10 accounts for the potentiation of the Sirt-6 signal.

Adiposity has well-documented consequences for insulin resistance [21]. As mentioned in Section 2, obesity induced by a 23-week high-fat and high-sucrose diet was mitigated in liver-specific *Usp20*KO mice, indicating that liver-specific *Usp20* deficiency caused significant restoration of glucose and insulin tolerance [50].

#### 3.2.5. Hypothalamus

Because the hypothalamus is the regulatory center of the autonomic nervous system and of glucoregulatory hormones, the activity of hypothalamic neurons influences circulating glucose levels. Dysregulation of UPS in the hypothalamic nuclei causes metabolic disorders in rodents [115], suggesting that hypothalamic USPs might contribute to blood glucose levels. In a transcriptomics study, experimental hypoglycemia increased the *Usp2* expression in several brain regions, including the hypothalamus [116]. Kitamura and colleagues recently showed that application of the USP2 inhibitor ML364 into the ventromedial hypothalamus (VMH) increased circulating norepinephrine levels and simultaneously increased glycogenolysis [117]. Consequently, blood glucose levels were substantially increased in the ML364-treated mice. Mechanistically, an ML364 treatment experiment suggests that USP2 inhibits the accumulation of ROS in VMH neurons, and thereby leads to continued ATP production in the mitochondria. As a result, low-level AMPK phosphorylation reduces the activation of the sympathetic nervous system [117]. Given that USP2 blockade–induced elevation of blood glucose levels occurs under normal feeding conditions, hypothalamic USP2 is likely necessary to maintain blood glucose levels at physiological concentrations.

## 4. Diabetes-Associated Disorders

In diabetic patients, the salient glycation of proteins and lipids produces AGEs [118]. AGEs bind to AGE receptors (RAGE), which provoke robust oxidative stress by activating several intracellular signaling molecules, including MAPKs, Janus kinases (JAKs), and NF-κB [118]. RAGE-elicited oxidative stress and local inflammation damage the microcirculation [118]. Additionally, high concentrations of glucose also cause oxidative damage in neural and Schwan cells due to the activation of the AGEs-RAGE system and insufficient removal of mitochondrial ROS [119,120]. Moreover, sorbitol, which is converted from glucose by aldose reductase, also leads to direct tissue damage and/or aberrant osmotic conditions in neural cells [119]. Based on these pathological characteristics, diabetic symptoms are often observed in the kidney (nephropathy), retina (retinopathy), peripheral skin (foot ulcers), and peripheral nerves (neuropathy). This section focuses on the role of USPs in these diabetes-associated disorders.

### 4.1. Nephropathy

Diabetic nephropathy is a form of chronic kidney failure that occurs in diabetic patients. The major symptoms of diabetic nephropathy are structural alterations in the glomeruli, decreased glomerular filtration rates, and proteinuria [121,122]. AGEs stimulate the accumulation of extracellular matrix (ECM) proteins, such as collagen and fibronectin, through the substantial induction of growth factors, including transforming growth factor-β (TGF-β) and connective tissue growth factor from mesangial cells [123,124]. Diabetic nephropathy is present in 20–50% of diabetic patients, and leads to end-stage renal disease requiring renal replacement therapy [125]. In most cases, diabetic nephropathy co-occurs with hypertension, which further exacerbates nephropathy [124,126]. Since the number of T2DM cases is increasing dramatically, the largest population of patients requiring dialysis consists of those with diabetic nephropathy [127]. Cumulative evidence suggests the participation of several renal USPs in diabetic nephropathy.

The progression of diabetic nephropathy is known to be controlled by sex steroid hormones [128]. For instance, ovariectomy promotes morphological changes in the glomeruli in diabetic rats, indicating that estrogen prevents the progression of diabetic nephropathy [129]. In a transcriptome analysis of renal cortex comparing diabetic mice with diabetic post-menopausal mice, *Usp2* expression was slightly enhanced in the menopause model [130]. By contrast, high-fructose corn syrup supplementation downregulated *Usp2* mRNA in the kidney via humoral factors derived from the liver [131]. Further studies are required to clarify the role of USP2 in the pathogenesis of diabetic nephropathy.

In addition to USP2, the expression of other USPs has also been shown to be regulated in the kidneys of T1DM rat models treated with the toxic glucose analogue STZ. After eight weeks of treatment with STZ, the expression of *Usp7*, *16*, *21*, and *22* was attenuated in the kidneys of diabetic rats [132]. Given that these USPs catalyze the ubiquitination of histones H2A and H2B [133], the authors speculated that downregulation of these USPs contributes to the epigenetic induction of ECM components [132]. Some studies have also highlighted the significant role of USP22 in diabetic nephropathy. AGEs blunted the expression of Sirt-1 in mesangial cells, leading to the aggravation of renal fibrosis [134]. Overexpression of *Usp22* sustained Sirt-1 levels in mesangial cells by avoiding the RAGE signal-induced proteasomal degradation of Sirt-1 [135]. In contrast, activation of AGE-RAGE signaling clearly decreased USP22 and Sirt-1 expression [135]. Therefore, USP22 signaling in mesangial cells has beneficial effects on the pathogenesis of diabetic nephropathy via Sirt-1 stabilization. AGE-stimulated downregulation of USP22 and Sirt-1 was also observed in rat proximal tubular epithelial cells [136]. Sirt-1 resists the AGE-induced expression of TGF-β signaling components, such as TGF-β1, TGF-β receptor, and Smad2/3, and inhibits fibronectin expression in tubular epithelial cells [134]. Therefore, USP22 might prevent tubular epithelial cells from adopting a pro-fibrogenic phenotype by stabilizing Sirt-1.

In contrast to the beneficial roles of USP22 in diabetic nephropathy, some reports have indicated the detrimental effects of USP22 in similar models. Mao et al. reported that *Usp22* mRNA expression in the kidneys was increased in STZ-treated rats [137]. The same authors also demonstrated that injection of *Usp22* siRNA rescued urine volume, urine protein to creatinine ratio, blood urea nitrogen, serum creatinine, and blood total cholesterol and triglyceride levels [137]. Moreover, *Usp22* silencing also suppressed renal cell apoptosis, decreased B-cell/CLL lymphoma 2 (Bcl-2)-associated X protein (*Bax*) mRNA, and concomitantly increased Bcl-2 [137]. Additionally, *Usp22* silencing increased the expression of ROS scavengers, leading to the mitigation of oxidative stress [137]. These results indicate that USP22 participates in the pathogenesis of diabetic nephropathy through oxidative stress-induced cellular damage. Similarly, Shi et al. showed the harmful effects of USP22 in podocytes [138]. Stimulation with high concentrations of D-glucose induced the expression of USP22 mRNA and protein in podocytes in a dose-dependent manner [138]. shRNA for *Usp22* also rescued high glucose stimulation–induced repression of William tumor-1 (WT-1) and synaptopodin, both of which are podocyte markers [138]. Moreover, *Usp22* knockdown abolished high glucose stimulation–induced cell cycle arrest, apoptosis, and proinflammatory cytokine production in podocytes [138]. In agreement with these findings, subcutaneous injection of *Usp22* shRNA-expressing lentivirus inhibited the increase in circulating proinflammatory cytokines and podocyte apoptosis in STZ-treated nude rats [138]. Further studies are required to address the discrepancies regarding USP22 function in diabetic nephropathy.

USP9X has been shown to inhibit the pathogenesis of diabetic nephropathy through multiple mechanisms. The USP9X protein was found to be expressed at lower levels in *db*/*db* T2DM mice compared to wild-type mice [139]. Accordingly, the USP9X protein was also downregulated in cultured renal epithelial-like NRK-52E cells exposed to high concentrations of glucose [139]. Overexpression of *Usp9x* in these cells attenuated glucose-induced epithelial-to-mesenchymal transition (EMT) in a DUB activity–dependent manner [140], which is a pathology associated with diabetic nephropathy [139]. In contrast, *Usp9x* knockdown enhanced the EMT transition following the induction of ECM components, such as fibronectin and collagens [139]. Mechanistically, USP9X directly stabilizes connexin 43 by removing polyubiquitin chains [139]. These results indicate that USP9X attenuates diabetic tubulointerstitial fibrosis by reducing the degradation of connexin 43. In addition to kidney epithelial cells, another report suggested a protective role of USP9X in mesangial cells [141]. In diabetic nephropathy, AGEs elicit the production of ECM components in mesangial cells via oxidative stress [142]. In diabetic models, ubiquitin-dependent degradation of nuclear factor erythroid 2-related factor 2 (Nrf2) is observed; Nrf2 is responsible for the transcription of genes encoding antioxidant proteins, such as heme oxygenase-1 (HOX1) and NAD(P)H:quinone oxidoreductase 1 (NQO1) [143]. Overexpression of *Usp9x* has been shown to stabilize Nrf2 and to increase *Hox1* and *Nqo1* transcription [141]. This agrees with the observation that *Usp9x* depletion substantially augmented ECM component production after AGE stimulation, whereas overexpression of *Usp9x* diminished AGE-induced ECM component production [141]. Therefore, USP9X represses ECM component production by increasing antioxidative activity in mesangial cells.

There are also reports that other USPs exacerbate diabetic nephropathy. Under diabetic conditions, the abundance of the USP36 protein increased in mouse kidneys and renal tubular epithelial cells (TECs) [144]. USP36 deubiquitinates the dedicator of cytokinesis 4 (DOCK4), as evidenced by the up- or downregulation of *Usp36* in TECs in response to overexpression or knockdown, respectively [144]. It is known that DOCK4 is required to maintain β-catenin expression, which participates in ECM component induction [145]. Accordingly, knockdown of either *Usp36* or *Dock4* attenuated β-catenin expression in conjunction with decreased production of ECM components [144]. Thus, USP36 worsens diabetic kidney fibrosis through the activation of the DOCK4-β-catenin system.

Podocyte injury is usually observed in diabetic nephropathy [146]. Like other USPs, USP14 in podocytes also plays a deteriorative role in the pathogenesis of diabetic nephropathy. Sperm-associated antigen 5 (SPAG5) was shown to be upregulated in cultured human podocytes treated with high concentrations of glucose [147]. Genetic silencing of *Spag5* expression suppressed the high glucose treatment–induced apoptosis of podocytes, and increased the levels of cleaved caspase 3, cleaved caspase 9, and BAX, but decreased Bcl-2 expression [147]. On the other hand, *Spag5* knockdown ameliorated high glucose-induced autophagy [147], which prevented podocyte damage [148]. In podocytes, SPAG-antisense 1 (SPAG5-AS1), a long non-coding RNA (lncRNA) transcribed closely from the *Spag5* gene, modulates SPAG5 by stabilizing its mRNA and protein [147]. For regulation at the protein level, the interaction of SPAG5-AS1 with USP14 is necessary for the accumulation of the SPAG5 protein in podocytes exposed to high glucose levels [147]. Interestingly, Akt, a downstream component of the SPAG5-elicited apoptosis signal, is known to activate USP14 through phosphorylation [149]. Hence, USP14, SPARC5-AS1, SPARC5, and Akt form a positive feedback loop that leads to the apoptosis of podocytes under diabetic conditions [147].

As with USP14, USP15 was also increased in podocytes following high glucose treatment [150]. Knockdown of *Usp15* mitigated apoptosis, oxidative stress, and NF-κB-dependent induction of proinflammatory cytokines elicited by high glucose treatment [150]. In agreement with this, USP15 deficiency also increased nuclear Nrf2 levels in conjunction with HOX1 and NQO1 expression [150]. Mechanistically, USP15 retains Nrf2 in the cytoplasm, stabilizing Keap1, an inhibitor of the nuclear translocation of Nrf2 [43,150].

### 4.2. Retinopathy

As with kidney blood vessels, hyperglycemia also damages capillary vessels that supply retinal cells. Although compensatory angiogenesis occurs in the retina, regenerated vessels are often leaky, and capillary cell types, such as pericytes and vascular smooth muscle cells, are lost at a later point [151]. Additionally, inflammatory cells accumulate around lesions and worsen damage to neural and glial cells. According to data from 288 studies of ~4 million participants from 98 different countries, 1.06% of cases of blindness and 1.3% of moderate or severe vision impairment among those aged 50 years and older were caused by diabetic retinopathy in 2015 [152]. Thus far, several USPs have been shown to impact the pathogenesis of diabetic retinopathy.

Many studies have demonstrated that excessive NF-κB–induced inflammatory responses are involved in retinal diseases, including diabetic retinopathy [153,154,155,156]. Among the proinflammatory cytokines, TNF-α provokes NF-κB activation in retinal pigment epithelial cells [157]. Although USP48 stabilizes NF-κB p65 in HeLa cells via the removal of the K48-linked polyubiquitin chain [158], gene silencing of *Usp48* increases nuclear NF-κB p65 protein in cultured retinal pigment epithelial cells without any stimulation [159]. After TNF-α stimulation, *Usp48* shRNA treatment sustained nuclear NF-κB p65 expression at a high level [159], and also increased the transcriptional activity of NF-κB in the retinal cells [159]. Considering that these data show preferential localization of USP48 in the nucleus of retinal cells, nuclear USP48 might mitigate retinal inflammatory responses by promoting the degradation of NF-κB p65 [159].

In contrast to USP48, there are reports that two USPs, namely USP1 and 14, are deteriorative in diabetic retinopathy. An anti-malarial drug, primaquine diphosphate (PD), interrupted vascular endothelial growth factor (VEGF)-induced conformational changes of actin fibers in vascular endothelial cells [160]. Consequently, PD blocked vascular leakage and attenuated the severity of diabetic retinopathy [160]. PD also blocked more than 50% of the DUB activity of USP1, but not USP2 [160]. Moreover, the *Usp1* shRNA conserved VEGF-induced permeabilization of vessels, implying that USP1 contributes to diabetic retinopathy.

Under conditions of long-term hyperglycemia, Müller cells, a glial cell population in the retina, exhibit induction of lncRNA-OGRU, which consists of 1816 ribonucleotides [161]. lncRNA-OGRU restored the expression of USP14 by interfering with micro RNA-320 (miR-320), which targets *Usp14* mRNA in Müller cells [162]. USP14 was shown to directly stabilize the TGF-β receptor and lead to the activation of TGF-β signaling, which is accelerated in Müller cells of diabetic nephropathy patients [162,163]. In addition, the increased expression of *Usp14* enhances inflammatory cytokine production in Müller cells in diabetic conditions via the activation of NF-κB signaling through increased IκBα degradation [162]. Moreover, the introduction of a *Usp14*-expressing construct increased ROS generation in Müller cells through marked suppression of superoxide dismutase activity and the repression of several antioxidant proteins [162]. USP14-evoked ROS accumulation is likely to elicit the digestion of Nrf2 by UPS [162]. Collectively, these data support the idea that USP14 aggravates diabetic retinopathy by enhancing inflammation and oxidative stress.

### 4.3. Neuropathy

In diabetic patients, metabolic dysfunction occurs in the peripheral nervous system, particularly in sensory axons and autonomic axons [120]. Occasionally, defects are apparent in motor axons [120,164]. Hyperglycemia causes a wide variety of changes, such as axonal degeneration, malfunction of Schwan cells, and accumulation of oxidative damage in the dorsal root ganglia [120]. In addition, ischemia caused by aberrant microcirculation is another source of damage to the peripheral nervous system [120]. One of the symptoms of diabetic neuropathy is chronic pain, which is caused by the continuous release of neurotransmitters, such as substance P, at afferent sensory nerves in the spinal dorsal horn [165]. Although several ion channels, such as Na_v_1.7, Na_v_1.8, and transient receptor potential cation channel subfamily V member 1 (TRPV1), have been shown to be responsible for diabetic pain, prolonged activation of a dominant voltage-gated T-type calcium channel, Cav3.2, is also considered to be a cause of diabetic neuropathic pain [120]. Cav3.2 has been demonstrated to accumulate in the dorsal horn and dorsal root ganglia following neural damage [166]. Since *Usp5* silencing dampened Cav3.2-dependent pain nociception, USP5 seems to sustain Cav3.2 protein expression in the dorsal horn [166]. *ob*/*ob* mice, which have T2DM due to leptin deficiency, suffer from hyperalgesia [167]. A small organic molecule called suramin greatly decreased diabetic neuropathy in *ob*/*ob* mice by causing the dissociation of Cav3.2 from USP5, indicating that USP5 promotes hyperalgesia [167]. Therefore, USP5 might be a therapeutic target to induce analgesia in neuropathy patients.

### 4.4. Myopathy

Skeletal muscle fails to maintain normal function under diabetic conditions [168]. Given that skeletal muscle plays a pivotal role in energy metabolism, physical activity, and myokine production [96,169], defects in this muscle group, such as sarcopenia, severely worsen the complications of metabolic disorders [170]. Diabetic myopathy is often accompanied by neuropathy, leading to further muscle weakness [164,171]. In severe cases, muscle atrophy or contracture is observed [172,173]. As previously mentioned, a prior study revealed the modulatory role of USP21 on metabolic changes in skeletal muscle in diabetic individuals [84]. Specifically, skeletal muscle-specific *Usp21*KO mice had increased soleus and tibialis anterior muscle mass, suggesting that USP21 is deteriorative for diabetes-induced muscle atrophy [84].

Treatment with STZ has been shown to gradually decrease extensor digitorum longus (EDL) muscle mass in rats [174]. Additionally, *Usp19* mRNA levels in the EDL muscle were negatively correlated with EDL mass [174]. These observations suggest that USP19 has the potential to sustain muscle mass under physiological conditions.

### 4.5. Cardiomyopathy

T2DM patients have an increased risk of several cardiac disorders, including myocardial infarction (MI) [175]. Interestingly, in T2DM models induced by a high-fat diet or leptin receptor deficiency, the left ventricular rejection fraction and left ventricular fraction shortening were decreased, while the ventricular internal diameter at end-diastole and the left ventricular internal diameter at end-systole were increased [176]. Moreover, additional treatment with MI caused further alterations in these indices [176]. In T2DM models, increases were found in α-smooth muscle actin, a myofibroblast marker, and the fibrosis markers collagen type I and matrix metalloprotease 9, suggesting the occurrence of cardiac fibrosis [176]. T2DM-induced cardiac fibrosis could be rescued by treatment with follistatin-like protein 1 (FSTL1), a cardiokine [176]. After FSTL1 treatment, USP10, and Notch1 receptor Noch1 intracellular domain 1 (NICD1) were increased in cardiac myocytes [176]. On the other hand, a USP10 inhibitor, spautin, and a Notch1 inhibitor, LY3039478, abolished the therapeutic effects of FSTL1 in myocytes under diabetic conditions [176]. Given that USP10 deubiquitinates NICD1 [177], USP10 seems to attenuate fibrosis through the activation of Noch1 after FSTL1 stimulation. In agreement with this finding, transfection with the *Usp10*-expressing plasmid reduced the expression of fibrosis markers, an oxidative stress marker, and a cellular damage marker in cardiomyocytes isolated from mice with T2DM [176]. Therefore, the FSTL1/USP10/Noch1 cascade plays a preventive role in diabetic cardiomyopathy.

### 4.6. Foot Ulcers

Foot ulcers are occasionally observed in diabetic patients due to impaired angiogenesis, malfunction of the peripheral nervous system, and subsequent bacterial infections [178]. A study that screened the expression of eight USPs found that USP7 expression was markedly increased in diabetic ulcer lesions relative to normal skin [179]. *Usp7* mRNA and USP7 protein were also upregulated by AGE stimulation in human umbilical vein endothelial cells (HUVECs) [179]. Conversely, knockdown of *Usp7* by shRNA dampened AGE-induced cell cycle arrest and cellular senescence in HUVECs, indicating that USP7 participates in AGE-induced dysfunction in veins [179]. Mechanistically, USP7 was shown to interact directly with p53, and augmented p53-p21 signaling in HUVECs [179]. Correspondingly, treatment with the USP7 inhibitor HBX41108 clearly inhibited STZ-induced foot ulcers in rats and decreased p53 and p21 expression [179]. Thus, USP7 interferes with vascular maintenance and is involved in diabetic ulcer formation.

## 5. NAFLD

NAFLD covers a wide pathological spectrum ranging from simple non-alcoholic fatty liver (NAFL) to non-alcoholic steatohepatitis (NASH). The latter represents more progressive symptoms, such as fibrosis and inflammatory cell infiltration into the liver, which are triggers for cirrhosis and hepatocellular carcinoma (HCC) [180]. Over the last two decades, NAFLD has become the most common chronic liver disease. According to a global epidemiological study, the estimated global prevalence of NAFLD is about one in four people [181]. Due to its similarity to metabolic disorders, there has been a proposal to rename NAFLD as metabolic dysfunction–associated fatty liver disease (MAFLD) [182]. NAFLD patients commonly suffer from obesity (51.3%), T2DM (22.5%), hyperlipidemia (69.2%), hypertension (39.3%), and metabolic syndrome (42.5%) [181]. Moreover, the NAFLD incidence rate is ~2-fold higher in T2DM patients [183], indicating that insulin resistance is an aggravating factor for NAFLD [182]. Interestingly, aberrant insulin signaling causes the induction of lipogenic genes through SREBP1c activation [182]. Dysregulation of lipid metabolism in the liver also negatively influences insulin sensitivity in energy metabolism–competent organs such as skeletal muscle and adipose tissue, which is followed by the progression of insulin resistance, or vice versa. Since NAFLD and T2DM have a common etiology, the restorative or deteriorative roles of hepatic USPs overlap in T2DM and NAFLD in many cases (Figure 2).

To date, several reports have demonstrated that USP7 modulates the pathogenesis of NAFLD and NASH [99,100,184,185]. Methionine adenosyltransferase (*Mat*)-deficient mice have been shown to have severe NASH and HCC due to *S*-adenosylmethionine–elicited activation of liver kinase B1 (LKB1) [186,187]. In agreement with these findings, hepatocytes lacking *Mat* or *Lkb1* underwent less apoptosis [185]. Mechanistically, LKB1 facilitates the dissociation of USP7 from murine double minute 2 (Mdm2), which is a p53-specific E3 ubiquitin ligase, and prevents p53-elicited cell death [185]. This observation suggests that USP7 is involved in the progression of mild NAFLD to NASH or HCC. Other studies have indicated that USP7 also contributes to lipid deposition in hepatocytes. Lee et al. reported that overexpression of *USP7* directly stabilizes PPARγ, a pivotal adipogenic transcription factor, in COS7 cells [100]. Additionally, *USP7* overexpression upregulated *PPARG* mRNA in HepG2 hepatocytes, which was followed by increases in glucose and fatty acid uptake [100]. Moreover, infection of *Usp7*-expressing adenovirus increased PPARγ levels in the liver of mice and accelerated the development of fatty liver [100]. Ni et al. also demonstrated the role of USP7 as a modulator of zinc finger protein 638 (ZNF638), which was previously shown to function as a positive transcriptional modulator of C/EBP β and C/EBP δ in adipocytes [188]. By employing overexpression and knockdown *Usp7* models, they demonstrated that USP7 increased ZNF638 protein expression in SK-Hep1 hepatocytes by two distinct mechanisms: direct stabilization by deubiquitination of ZNF638, and induction of *ZNF638* mRNA by stabilization of cAMP-responsive element binding protein (CREB), an upstream regulator [184]. Furthermore, USP7 activity increased the levels of the cleaved form of SREBP1c in the nucleus by deubiquitination. Consequently, the USP7-ZNF638-SREBP1c complex activated the transcription of lipogenic genes encoding FASN, acetyl-CoA carboxylase (ACC), and stearol-CoA desaturase [184].

Paradoxically, the opposite effect of USP7 on NAFLD has also been documented. As mentioned in Section 3.2.4, PiT1 tethers USP7 to switch off insulin signaling by promoting IRS1 degradation [99]. Thus, the lack of PiT1 in hepatocytes dramatically improved insulin sensitivity in mice due to the interplay between USP7 and IRS1 [99]. Liver weights in these mice decreased by almost 50% and exhibited a dramatic decrease in hepatic triglyceride content [99]. If the interaction between USP7 and IRS1 improves insulin sensitivity in the liver, then USP7 mitigates NAFLD progression. Further investigation is needed to uncover the detailed role of hepatic USP7 in the incidence and progression of NAFLD.

As noted, hepatic USP4, 10, and 18 have preventive roles in the pathogenesis of diabetes mellitus [109,111,114]. These USPs in hepatocytes also had a protective effect against NAFLD induced by a high-fat diet or leptin deficiency [109,111,114]. For instance, infection with a *Usp4*-expressing adenovirus vector decreased liver weight, hepatic triglycerides, total cholesterol, and non-esterified fatty acids in *ob*/*ob* mice [109]. Moreover, hepatic steatosis was ameliorated in liver-specific *Usp4* transgenic mice but worsened in liver-specific *Usp4*KO mice [109]. Abundance of USP4 in the liver substantially influenced the expression of genes participating in cholesterol efflux, fatty acid uptake, fatty acid synthesis, and β-oxidation [109]. The lack or overexpression of the *Usp4* gene in hepatocytes increased or decreased serum liver damage markers, including aspartate aminotransferase and alanine aminotransferase, respectively. [109]. Furthermore, NF-κB–dependent induction of proinflammatory cytokines was up- and downregulated in the livers of *Usp4*KO and *Usp4* transgenic mice, respectively [109]. Likewise, liver-specific transgenic or knockout mice for *Usp10* or *Usp18* displayed mitigation or progression of hepatic steatosis, along with changes in the expression of proinflammatory cytokines and lipid-metabolizing enzymes [111,114]. In agreement with these data, treatment with *Usp10* or *Usp18*-expressing adenovirus drastically improved NAFLD in *ob*/*ob* mice or high fat diet–fed mice [111,114]. The therapeutic effects of USP10 on NAFLD have also been documented in a report using another NAFLD model. Since methionine and choline are required for hepatic β-oxidation and the synthesis of very low-density lipoprotein (VLDL), methionine, and choline-deficient diets (MCDD) dampen the release of lipids like VLDL from the liver to the circulation [189,190]. Consequently, the liver becomes burdened with excessive lipid deposits, massive oxidative stress, and local inflammation, eventually leading to NASH [189,190]. Relative to high-fat diets, MCDD causes short-term histological changes in the liver of rodents, but does not cause hyperlipidemia and insulin resistance [190]. Using MCDD-fed mice as a NASH model, Xin et al. found a positive correlation between the expression of *Usp10* and autophagy components, such as microtubule-associated protein light chain 3B-1 (*Lc3b*), and autophagy target gene 5 (*Atg5*) and 6 (*Atg6*) [191]. Treatment with *Usp10-*expressing adeno-associated viral particles activated the autophagy pathway, and improved fibrosis and lipid accumulation in the liver of MCDD-fed mice [191]. Moreover, chloroquine or 3-methylaenine, inhibitors of autophagy, abrogated the suppressive effects of USP10 on lipid accumulation in hepatocytes [191]. Thus, USP10 also has the potential to overcome lipotoxicity through the activation of autophagy in the liver.

Using the Cancer Genome Atlas database, Chen et al. found that the expression of some USPs was closely linked to a poor prognosis in cancer [192]. Among the USPs found, a high expression of USP22 was observed in HCC, and its expression was closely associated with the increase of lipids and lipid-like metabolites [193]. Further gene set enrichment analyses using the same database also suggested that USP22 contributes to unsaturated fatty acid biosynthesis in HCC [193]. Given that excessive de novo lipogenesis is proposed to contribute to the pathological progression from NASH to HCC [194], USP22 is likely to be a key molecule in the transition to cancer in hepatocytes. In support of this, gain-of-function or loss-of-function studies revealed that USP22 stimulates lipogenesis from glucose in cultured hepatocytes and subsequently contributes to tumorigenesis [193]. USP22 directly stabilizes PPARγ through deubiquitination, resulting in the upregulation of pivotal lipogenic enzymes, including ACC and ATP citrate lyase. This suggests that USP22 accelerates the transformation from NAFLD to HCC by enhancing lipogenesis in hepatocytes. In contrast to the harmful effects of USP22 in late phases of NAFLD, another study implied that USP22 attenuates lipid deposition in the liver by modifying mitochondrial respiration. G protein α12 inhibits fasting-induced hepatic steatosis by inducing the accumulation of Sirt-1, PGC1α, and carnitine palmitoyl transferase-1, all of which are associated with mitochondrial function [195]. In line with this, *Gna12* (the gene encoding G protein α12) KO mice display aberrant oxygen consumption and β-oxidation [195]. Because treatment with siRNA for *USP22* diminished G protein α12–induced accumulation of Sirt-1 in HepG2 cells, it was concluded that UPS22 stabilizes Sirt-1 [195]. G protein α12 elevates USP22 expression in hepatocytes by activating hypoxia-inducible factor 1α (HIF1α) [195]. Therefore, USP22 regulates the suppressive role of G protein α12 on NAFL through the upregulation of mitochondrial activity via Sirt-1 stabilization. Further studies might be required to address the discrepancy regarding the roles of USP22 on lipid deposition in hepatocytes.

Previous reports demonstrated that USP2 and 14 deubiquitinate, thereby increasing FASN [48,108], which synthesizes palmitates from acetyl-CoA and malonyl-CoA in the presence of nicotinamide adenine dinucleotide phosphate (NADPH) [196]. In addition tobeing a source of lipid droplets in hepatocytes, palmitate also contributes to insulin insensitivity and the activation of apoptosis in hepatocytes [197]. Therefore, USP2 and 14 appear to contribute to the progression of NAFLD. USP2 interacts with FASN in the mouse liver, human hepatic cancer, and HepG2 cells, and subsequently prevents the ubiquitination-dependent digestion of FASN [198,199]. Conversely, binding O-linked N-acetylglucosamine moieties to FASN interrupts USP2-FASN interactions, followed by attenuation of FASN-mediated lipogenesis in the liver [198]. On the other hand, infection of mice with a *Usp14*-encoding adenovirus via the tail vain substantially enhanced triglyceride content in the liver and elevated plasma triglyceride levels [108]. Since shRNA against *Fasn* mRNA abrogated *Usp14*-induced hypertriglycemia, stabilization of FASN by USP14 accounts for hepatosteatosis [108]. In agreement with these findings, *Usp14* silencing in mice clearly improved steatosis and decreased hepatic FASN expression [108].

As described in Section 2, hepatic USP20 determines the rate of cholesterol biosynthesis by stabilizing HMGCR [50]. Thus, hepatic USP20 influences not only the fat mass of adipose tissue but also triglyceride content in the liver. While wild-type littermates exhibited severe steatosis after 23 weeks of high-fat and high-fructose diets, liver-selective *Usp20*KO mice had dramatically improved hepatic lesions [50]. Moreover, daily injection of a chemical inhibitor of USP20 for two weeks significantly decreased the triglyceride content of the liver and serum, and improved steatosis [50]. Thus, hepatic USP20 is a complicating factor for NAFLD progression.

Krüppel-like factor 4 (KLF4) has been postulated to be a tumor suppressor of HCC [200]. USP11 interacts with KLF4 and digests the K63-linked polyubiquitin chain on KLF4 [201]. Gain-of-function and loss-of-function experiments revealed that USP11 facilitated the proliferation of hepatocytes and accelerated their transition towards cancer by promoting KLF4 instability [201]. Additionally, a mixture of oleate and palmitate decreased KLF4 mRNA and protein expression but increased USP11 in HepG2 cells, suggesting that USP11 might also be involved in the pathogenesis of NAFLD [201]. In support of this, shRNA against *Usp11* mRNA downregulated the expression of genes involved in fatty acid uptake and fatty acid synthesis, but upregulated genes promoting fatty acid β-oxidation, and reduced intracellular triglyceride content [201]. Furthermore, the mRNA levels of *Klf4* and *Usp11* were negatively correlated in NAFLD patients [201]. In summary, USP11 contributes to not only oncogenesis of HCC but also the progression of NAFLD.

Since cholesterol is crucial for cellular proliferation, supplementation with cholesterol esters triggers aggressive phenotypes in several cancers [202,203]. Correspondingly, sterol O-acetyltransferase (SOAT) 1, an enzyme that synthesizes cholesterol ester, promotes the proliferation and migration of HCC, indicating that the malignancy of HCC is also correlated with cholesterol content [204]. Additionally, high-fat and high-cholesterol diets triggered hepatocarcinogenesis in *p53*-deficient mice, indicating that the accumulation of cholesterol also triggers oncogenesis in hepatocytes [205]. The *p53*KO mice fed a high-fat and high-cholesterol diet displayed aberrant regulation of SOAT1 at both the transcriptional and post-transcriptional levels [205]. USP19 is among 95 known DUBs that interact with SORT1 in the context of post-transcriptional control. Experiments using models of *USP19* overexpression and knockdown demonstrated that USP19 decreased K33/K48-linked ubiquitination at the 120th lysine in SORT1 and prevented its digestion by the proteasome in hepatoma cell lines [205]. p53 directly attenuated the transcription of the *Usp19* gene by binding to its promoter [205], and the hepatocarcinogenesis induced by high-fat and high-cholesterol diets in *p53*-deficient mice was rescued in *Usp19*KO mice [205]. Thus, USP19 contributes to the initial phase of Western diet–induced HCC by modifying cholesterol biosynthesis.

## 6. Atherosclerosis

Western diets increase LDL cholesterol levels and decrease high-density lipoprotein (HDL) cholesterol levels in the blood [206]. Several metabolic factors, including hyperglycemia and hypertension, cause injury to vascular endothelial cells and facilitate the perfusion of blood LDL into the intima of arteries. Immersed LDL is then oxidized by oxidative stress in the area and is taken up by monocyte-derived macrophages or smooth muscle cells [207,208,209,210]. Subsequently, lipid-laden macrophages (and/or macrophage-like smooth muscle cells), called foam cells, form the lipid core of atherosclerotic plaques [210]. The narrowing of blood vessels by plaques raises local blood pressure and causes insufficient blood supply [211]. Moreover, local inflammation near the plaque elevates the fragility of vessel walls, occasionally leading to a blood vessel rupture, and subsequently cardiac vascular disease and stroke [210,211]. Several USPs have been shown to determine the pathological progression of atherosclerosis (Figure 3).

Surface expression of the LDL receptor (LDLR) in the liver is one of the determinants of blood LDL levels [212]. The inducible degrader of LDLR (IDOL) is an E3-ubiquitin ligase that selectively catalyzes the ubiquitination of LDLR [213]. Since overexpression of *Usp2* promotes the deubiquitination of IDOL, USP2 was considered to suppress LDL uptake via ubiquitin-dependent degradation of LDL [212]. However, the USP2-induced deubiquitination of IDOL deterred the proteasomal degradation of LDLR, but prolonged the half-life of IDOL [212]. Transfection of a *Usp2*-expressing construct restored IDOL-repressed LDL uptake in cultured cells, whereas *Usp2* knockdown reduced LDL uptake by ~50% in HepG2 hepatocytes [212]. These results suggest that USP2 promotes LDL uptake from circulation to hepatocytes. Since blood LDL levels are a prerequisite for atherogenesis, USP2 seems to act as a protective enzyme against atherosclerosis.

USP20 was demonstrated to be a positive regulator of cholesterol synthesis in the liver after feeding [50]. In agreement with this, liver-specific *Usp20* deficiency decreased serum LDL cholesterol and LDL triglyceride levels in mice and also attenuated HDL and VLDL levels [50]. Considering the pathogenetic roles of LDL, hepatic USP20 might contribute to the exacerbation of atherosclerosis. In sharp contrast, USP20 in vascular smooth muscle cells has an opposite effect on disease progression. Since TRAF6 is required for the activation of Toll-like receptor 4 (TLR4)-elicited NF-κB signals [214], TRAF6 may contribute to vascular inflammation. USP20 has been shown to form a complex with β-arrestin 2 and TRAF6 and to promote TRAF6 deubiquitination [215]. In addition, the introduction of an enzymatically inactive mutant of *Usp20* substantially increased the phosphorylation of the NF-κBp65 subunit and upregulated the expression of vascular cell adhesion molecule 1 (VCAM-1) in mouse carotid arteries [215]. In contrast, NF-κBp65 was significantly repressed, and VCAM-1 was decreased in cultured vascular smooth muscle cells overexpressing *Usp20* in response to challenge with bacterial lipopolysaccharide (LPS), the TLR4 ligand [215]. Considering that vascular inflammation contributes to the pathogenesis of atherosclerosis, USP20 in smooth muscle cells is likely to attenuate the incidence of atherosclerosis. Similarly, overexpression of the dominant negative mutant of USP20 in smooth muscle cells increased the phosphorylation of NF-κB p65 and expression of VCAM-1 in LDL receptor (*Ldlr*) KO mice, which have markedly high serum LDL levels [216]. Moreover, ectopic expression of *Usp20* in smooth muscle cells significantly reduced aortic atherosclerosis in *Ldlr*KO mice, while the introduction of the dominant negative form of USP20 increased atherosclerotic plaques [216]. Co-immunoprecipitation showed that USP20 interacts with several NF-κB signaling components, including receptor-interacting protein kinase 1 (RIPK1) [216]. At the same time, *Usp20* silencing strongly augmented NF-κB activation in response to stimulation with TNF-α or IL-1β [216]. Thus, USP20 in vascular smooth muscle cells mitigates atherosclerosis by repressing NF-κB-evoked inflammatory responses.

The role of USP14 in atherosclerosis is currently under debate. Liu et al. reported that the mRNA and protein levels of USP14 were, respectively, ~4- and ~3-fold more abundant in atherosclerotic tissues than in healthy ones [217]. In cultured human aortic smooth muscle cells, USP14 expression was upregulated by stimulation with platelet-derived growth factor-BB (PDGF-BB), which is primarily released from platelets and endothelial cells in damaged blood vessels [217]. Genetic silencing of *Usp14* restored the PDGF-BB-attenuated expression of smooth cell signatures, such as α-smooth muscle actin, calponin, and smooth muscle myosin heavy chain [217]. Additionally, *Usp14* knockdown also restored the phosphorylation of mammalian targets of rapamycin (mTOR) and S6 kinase that was induced by PDGF-BB treatment [217]. The mTOR activator MHY1485 reversed the effects of *Usp14* deficiency on PDGF-BB-stimulated smooth muscle cells [217]. Because USP14 binds to the 26S proteasome non-ATPase regulatory subunit (PSMD) 7 [217], and PMSD7 affects mTOR signaling [218], a complex involving USP14 and PMSD7 therefore mediates PDGF-BB-stimulated mTOR signaling. This complex is proposed to trigger the de-differentiation of smooth muscle cells from a quiescent state to a proliferative and migratory phenotype [219]. Given that smooth muscle cells exert this phenotypic switch in atherosclerotic lesions [220], USP14 in smooth muscle cells appears to trigger atherosclerosis. In contrast to the function of USP14 in smooth muscle, Fu et al. reported that the expression of *Usp14* was significantly downregulated in oxidized LDL-stimulated endothelial cells [221]. Moreover, overexpression of *Usp14* in endothelial cells interfered with NF-κB activation by deubiquitinating the NOD-like receptor family cascade recruitment domain family domain containing 5 (NLRC5) [221], which inhibits NF-κB activation [222]. Furthermore, infection with *Usp14*-expressing adenovirus halted the progression of atherosclerosis in apolipoprotein E (*Apoe*) KO mice [221]. Therefore, endothelial USP14 plays a suppressive role in inflammation of atherosclerotic lesions.

From investigations using siRNA libraries of DUBs, USP9X was identified as a DUB that prevents oxidative LDL uptake by THP-1 cells, a macrophage cell line [223]. The expression of USP9X was downregulated in macrophages in atherosclerotic lesions of human patients and a mouse model [223]. Macrophage-selective *Usp9x*/*ApoE* double KO mice had enhanced foam cell formation compared with *ApoE*KO mice, although there were no obvious changes in plasma cholesterol levels, triglyceride levels, or LDL and HDL levels [223]. Overexpression and gene interference experiments indicated that USP9X digests the K63 polyubiquitin chain at the K27 residue of class A1 scavenger receptor (SR-A1), but not other scavenger receptors in cultured cells [223]. K63 polyubiquitination on SR-A1 stimulated oxidized LDL uptake and proinflammatory cytokine expression in macrophages [223], suggesting the protective roles of macrophage USP9X for atherosclerosis. As expected, the administration of an inhibitory peptide against USP9X for 10 weeks promoted foam cell formation and atherosclerosis in *ApoE*KO mice [223]. Therefore, the retention of USP9X on SR-A1 in macrophages is likely to be a promising therapeutic strategy for atherosclerosis treatment.

In a cohort study of 1004 atherosclerotic patients, obesity was positively associated with plasma levels of wingless-related integration site 5A (Wnt5A), which is released from adipose tissue [224]. The same study found that the expression of Wnt5A receptors Frizzled2 and Frizzled5, but not Ror1, increased in the arterial walls of atherosclerotic patients [224]. Wnt5A strengthens NADPH-stimulated ROS production, primarily in the smooth muscle cell layer of human internal mammary arteries and mouse aorta [224]. Wnt5A-induced ROS generation is dependent on RAS-related C3 botulinus toxin substrate 1 (RAC1) activation and the subsequent membrane translocation of RAC1 and the NADPH oxidase P47^phox^ subunit [224]. Moreover, Wnt5A triggers the migration of vascular smooth muscle cells in a redox-dependent manner [224]. Analysis of transcriptomic data of vascular smooth muscle cells treated with Wnt5A alone, or with Wnt5A and an intracellular superoxide scavenger, revealed that the expression of *Usp17* was positively correlated with oxidative stress [224]. Genetic silencing of *Usp17* abolished the formation of GTP-bound RAC1 after Wnt5A stimulation in HeLa cells [224]. Since vascular smooth muscle cell migration is believed to be crucial for the early stages of atherogenesis, USP17 in smooth muscle cells might be required for the pathogenesis of atherosclerosis triggered by adipose-derived Wnt5A.

## 7. Cushing Disease

Presently, knowledge about the influence of somatic mutations in USP genes in metabolic disorders is limited. However, the involvement of somatic mutations in USP genes in Cushing disease has been relatively well investigated. Cushing disease is a metabolic disease that is attributed to extremely high levels of blood glucocorticoids [225]. Due to the hyperactivity of these glucocorticoids, patients suffer from adiposity and hyperglycemia [225]. Cumulative evidence indicates that genetic mutations in *USP8* loci cause Cushing disease in a large proportion of patients [226,227,228,229]. An early study detected somatic mutations of *USP8* genes in 4 out of 10 corticotroph adenomas [229], while another study showed that 36% of 134 functioning corticotroph adenomas had mutations in the *USP8* gene [226]. A Sanger sequencing study of adrenocorticotropic hormone (ACTH)-secreting adenomas similarly observed 17 types of *USP8* variants in 67 of 108 (~62%) ACTH-secreting tumor cells [230]. All these reports demonstrated that the mutations are present between amino acids 713 and 720 in exon 14, where the 14-3-3 protein binds [226,229,230]. These mutations conferred stronger enzymatic activity to USP8 and promoted the deubiquitination of the epidermal growth factor receptor, which resulted in excessive activation of the promoter for the proopiomelanocortin (*POMC*) gene, which encodes a precursor of ACTH [226,229,230]. In a more recent study, the *POMC* gene was shown to have two promoters located upstream of exon 1 and near the intron/exon 3 boundary [231]. Since corticotroph adenomas harboring *USP8* mutations are scarcely methylated at the first promoter, adenomas of Cushing disease are likely to utilize the first promoter to transcribe the *POMC* gene.

The influence of *USP8* mutations on some key molecules in the ACTH secretion process has also been assessed [232]. In *USP8* mutated tumors, the cell cycle regulator p27/kip was significantly decreased, and CREB phosphorylation was augmented [232]. Moreover, transfection with mutated *USP8* genes exclusively increased phosphorylated CREB and the nuclear expression of the PKA Cα subunit, indicating that the USP8 mutation activates PKA-CREB signaling [232]. Additionally, *USP8* mutations also increased the expression of heat shock protein 90, a chaperone protein for the glucocorticoid receptor [232]. Another report revealed that RA-9, a *USP8* inhibitor, impedes *USP8* mutation-evoked increases in ACTH secretion [233]. RA-9 treatment suppressed the phosphorylation of extracellular signal-regulated kinase (ERK) and CREB by ~50% and increased p27 expression by ~150% [233].

There is also a report demonstrating an association between USP8 and canine ACTH-secreting adenomas [234]. Sanger sequencing of 38 canine ACTH-secreting adenomas failed to detect somatic mutations in the *USP8* loci [234]. However, a population that showed preferential localization of USP8 in the nucleus had high ACTH production, even though the tumor sizes were small [234]. Together with the fact that mutated USP8 is localized to the nucleus in patients with Cushing disease [226,230], it was concluded that the substantial production of ACTH from corticotroph tumors might require nuclear localization of USP8.

Adrenalectomy is a therapeutic strategy for corticotroph tumors. However, refractory tumors, termed Nelson’s tumors, occasionally develop in bilateral adrenalectomized patients [235]. A retrospective study evaluating somatic mutations in the *USP8* gene in Nelson’s tumors indicated that 15 of 33 (45%) samples had functional mutations in exon 14 of the *USP8* gene [236]. Patients with mutations in the *USP8* gene did not display changes in body mass index, hyperpigmentation, size of tumors, age at diagnosis of Nelson’s tumor, or mortality [236], but had higher levels of ACTH in their blood after surgery when compared to patients without *USP8* mutations [236].

Somatic mutations in the *USP48* gene have also been found but have a lower contribution to the incidence of Cushing disease than the *USP8* gene. Exosome sequencing of 18 *USP8* non-mutated corticotroph adenomas (Cushing disease, *n* = 15; Nelson’s tumors, *n* = 3) indicated that the prevalence of USP48 mutations was 17% (3 samples) [237]. Additional analysis of 175 Cushing disease tumors showed that 9.1% (16 samples) had somatic mutations in the *USP48* locus [237]. Compared with wild-type USP48, mutated USP48 had an altered conformation of the catalytic subunit that contributed to its enhanced activity [237]. In agreement with these findings, *USP48* mutations induced hyper-release of ACTH, which resulted in increased expression of the *POMC* gene in AtT cells after stimulation with corticotrophin-releasing hormone [237].

## 8. Perspectives

Several papers have demonstrated that USPs are stabilizers of molecules that critically regulate energy metabolism. That is to say, USP7 stabilizes FoxO1, IRS1, SREBP1c, and PPARγ [45,99,101,184,238]; USP22 stabilizes Sirt1 and PPARγ [135,136,193]; and USP2 stabilizes FASN, IDOL, and C/EBPα [37,102,212]. Although these studies uncovered the pathological significance of USPs in metabolic disorders, critical issues still remain to be addressed. One of the major issues is assessing the contribution of each USP to metabolic disorders. Specifically, USP4 and 18 share TAK1 as a direct target but do not compensate for each other’s functions [109,111]. Thus, the level of overlap in the function of these USPs for hepatic homeostasis and the pathogenesis of NAFLD needs to be uncovered. There are also reports showing that USP4, 7, 10, 18, and 22 attenuate NAFLD progression [98,108,110,113,194,198], whereas other reports have indicated that USP2, 7, 11, 14, 19, 20, and 22 exacerbate NAFLD [49,99,107,187,188,196,201,202,204,207]. It is presently not possible to create a progress chart of NAFLD, including all USPs listed above. Similarly, independent studies identified USP36 in renal epithelial cells, and USP14, 15, and 22 in podocytes as complicating factors for diabetic nephropathy; however, the contribution of each USP to disease etiology is still unclear [138,144,147,150]. Combinatorial KOs of several USPs and the concomitant transduction of USPs might clarify the role of each USP in metabolic disorders and downstream molecular events. In parallel with this, comprehensive monitoring of USPs by OMICS approaches might not only identify novel USPs associated with metabolic disorders, but also reveal the dynamics of USP networks in the course of metabolic diseases like cancer [192].

The roles of USPs that are simultaneously expressed in different tissues must be taken into consideration. In hepatocytes, USP2 increases blood glucose levels via gluconeogenesis, which suggests that it plays a pathogenic role in the progression of diabetes mellitus [102]. Conversely, hypothalamic USP2 alleviates hyperglycemia, indicating anti-diabetic roles [117]. Moreover, macrophage USP2 represses adipose tissue inflammation and subsequent insulin resistance [79]. Thus, the impact of USP2 on metabolic disorders seems to be dependent on the cell types expressing it. In some cases, the functional diversity of USPs is evident between cell types in the same organ. USP22 in podocytes promotes diabetic nephropathy by enhancing oxidative stress [138], while USP22 in mesangial cells suppresses fibrosis through the stabilization of Sirt-1 [135]. Likewise, USP20 in hepatocytes and USP14 in smooth muscle cells exacerbate atherosclerosis, whereas USP20 in smooth muscle cells and USP14 in endothelial cells improve disease outcomes [215,216,217,222]. The contribution of each USP in each tissue should be compared to clarify avenues for therapeutic intervention. Alternatively, cell-type-specific drug delivery systems should be developed to selectively inhibit certain USPs in target cells.

Since a large proportion of USPs participate in carcinogenesis, various chemical inhibitors of USPs have recently been developed as candidates for anti-cancer drugs [30,32,33,239]. These inhibitors also enable us to analyze the pathophysiological roles of each USP in the context of other diseases, including metabolic disorders. Several studies have utilized chemical blockers to assess the roles of USP in metabolic disorders; e.g., GSK2643953A for USP20 in obesity, ML323 and SJB-043 for USP1 in β-cell damage, ML364 for USP2 in hypothalamic function, and HBX41108 for USP7 in diabetic foot ulcers [50,69,117,179]. USP inhibitors are especially useful for the assessment of the biological function of USPs, which can be compensated for by other USPs in long-term experiments. On the other hand, we must take into consideration the specificity of each inhibitor. Most USP inhibitors have been validated for their specificity using a limited number of USP species [33]. Considering that even common inhibitors have unexpected effects on different enzymes [240,241], gathering information about the specificity of USP inhibitors is desirable.

Although many papers have demonstrated that the regulatory roles of USPs via the stabilization of target proteins determine the incidence of metabolic diseases, the dynamics of the interaction between USPs and the proteasome in metabolic diseases are not fully understood. USP14 is the only USP protein that reversibly associates with the 19S regulatory particle of the proteasome and promotes its maturation [242,243,244], suggesting that USP14 intrinsically controls ubiquitination of target protein as well as proteasomal activity [245]. Full-length USP14, or the UBL domain of USP14, blocks target digestion by the proteasome via an allosteric mechanism [245], whereas the ligation of an ubiquitylated target to USP14 potentiates proteasomal ATPase activity and 20S core particle opening, followed by proteasomal digestion [245]. With respect to the modulation of key molecules for metabolic diseases, such as FASN and CBP [103,221], USP14 might not only disassemble the ubiquitin chain on target proteins, but might also affect the incorporation of the targets into the proteasome. Interestingly, USP14 binds PSMD7, a regulatory subunit of the 19S proteasome [217]. Knockdown of *PSMD7* dramatically decreased proteasome activity and repressed mTOR/p70S6 kinase [218]. Furthermore, *Usp14* knockdown attenuated phosphorylation of both mTOR and p70S6 kinase after PDGF-BB stimulation [217]. mTOR affects cellular metabolism by various pathways, including protein synthesis, lipid synthesis, and energy metabolism [246]. Given that changes in mTOR activity also influence the incidence of metabolic disease at an individual level [246], regulatory links between the proteasome and mTOR might be therapeutic targets for metabolic disorders. In this respect, further studies investigating the roles of USPs in UPS are required.

Hyperactive mutations of *Usp8* and *Usp48* cause excessive ACTH secretion from corticotroph adenomas, implying that USP8 and 48 have the potential to modulate the hypothalamus-pituitary-adrenal axis under physiological conditions. Correspondingly, changes in *Usp4*, *10*, and *18* expression in NAFLD suggest that these USPs putatively maintain tissue homeostasis under physiological conditions; however, their respective KO models did not elicit hepatic changes under normal feeding conditions [109,111,114]. On the other hand, the blockade of USP2 or 19 dramatically affected blood glucose levels [102,117] and fat mass [49] under normal feeding conditions, indicating that they play regulatory roles under physiological conditions. Further clarification of USP targets might shed new light on novel rate-limiting steps for physiological events. Since healthy homeostatic conditions lower the incidence of metabolic disorders, modification of the interplay between USPs and their targets might lead to both therapeutic and preventive strategies.

This review article focuses on the contributions of USPs to metabolic disorders at the individual level. At the cellular level, USPs might control a wider variety of cellular functions by modifying energy metabolism. For instance, the *Usp2*KO models displayed defects in male subfertility, presumably by aberrant ATP production in sperm [247]. Accordingly, USP2 in testicular macrophages maintains spermatic ATP synthesis by stimulating granulocyte macrophage-colony stimulating factor (GM-CSF) secretion from macrophages [248]. *Usp2* deficiency–induced subfertility can thus be regarded as a metabolic disorder in a broader sense of the term, although it does not manifest as a disruption of energy metabolism at the individual level. In addition, USP2 can be regarded as a regulator of cellular energy metabolism in myoblasts, since USP2 maintains ATP supply by protecting mitochondria from ROS [249]. Thus, USP2 maintains energy metabolism at a cellular level. Interestingly, ROS-induced mitochondrial dysfunction is attributable to metabolic disorders at the individual level [15,250,251], suggesting that USP2 might affect energy consumption more broadly. Further studies might clarify the causal relationship between USP-controlled energy metabolism and disorders at the cellular and individual levels.

## 9. Short Summary and Conclusions

The findings detailed in this review are summarized in Table 1. Since USP2, 7, 15, 19, and 20 stimulate either adipogenesis or lipid synthesis, they are likely to cause obesity, while USP53 might be restorative for obesity. Under T2DM conditions, several USPs have positive or negative effects on energy metabolism–competent organs. In the pancreas, USP22 protects ROS-induced β-cell death, while USP1 participates in DNA-damage-induced β-cell apoptosis. In skeletal muscle, USP21 aggravates glucose metabolism by interfering with mitochondrial genesis, while USP9X, 20, and 33 restore muscular insulin signaling. Hepatic gluconeogenesis is one of the determinants of circulating glucose levels. USP2 and 14 enhance gluconeogenesis, leading to high blood glucose levels. In contrast, USP7 inhibits hyperglycemia by mitigating gluconeogenesis. Additionally, USP7 improves the insulin sensitivity of the liver. Likewise, hepatic USP4, 10, and 18 restore insulin resistance, whereas USP14 and 20 decrease it. USP2 in the ventromedial hypothalamus alleviates sympathetic nervous activation, resulting in the attenuation of glycogenolysis. USPs also modulate diabetes-induced complications. USP9X in renal epithelial cells, and USP22 in renal epithelial cells and mesangial cells have beneficial effects on diabetic nephropathy. However, USP14, 15, and 22 in podocytes and USP36 in epithelial cells have adverse effects. USP1 in retinal vascular endothelial cells and USP14 in Müller cells are complicating factors for diabetic retinopathy, whereas USP28 in retinal pigment epithelial cells suppresses the progression of retinopathy. USP5, 7, and 10 are involved in the pathogenesis of diabetic neuropathy, cardiomyopathy, and foot ulcers, respectively. Additionally, USP19 and 21 have been proposed to participate in diabetic myopathy. USP4, 10, and 18 mitigate the progression of NAFLD, but USP2, 11, 14, 19, and 20 exacerbate it at different stages. The pathophysiological roles of USP7 and USP22 in hepatocytes in the context of NAFLD remain controversial. Regarding atherosclerosis, USP20 in smooth muscle cells has a restorative role characterized by the modulation of inflammatory signals, but it increases risk by promoting cholesterol synthesis in the liver. USP14 also has diverse influences on atherosclerosis development, depending on the cells that express it: USP14 in smooth muscle cells inhibits atherosclerosis, and USP14 in epithelial cells represses atherosclerotic inflammation. USP17 increases the risk of atherosclerosis by promoting the accumulation of oxidative stress, whereas USP2 and 9X hinder the progression of disease by modulating cholesterol uptake in the liver and macrophages. USP18 and 22 are considered to be involved in the incidence of T1DM and lifestyle-related diseases. Moreover, somatic mutations in the *Usp8* and *Usp48* genes have frequently been observed in patients with Cushing syndrome, suggesting regulatory roles of these USPs in energy metabolism. Taken together, USPs seem to be pivotal modulators of metabolic disorders. Due to the dependence of their biological function on DUB enzymatic activity, chemicals targeting USPs can be designed. USP-targeting therapy may be applied to metabolic disorders in the future, but establishing these therapies requires overcoming issues related to specificity and tissue selectivity for each chemical.

Metabolic disorders, USP species, USP-expressing cells or tissues, pathological roles of USPs, direct molecular targets, and references are shown.

## Figures and Tables

**Figure 1 ijms-24-03219-f001:**
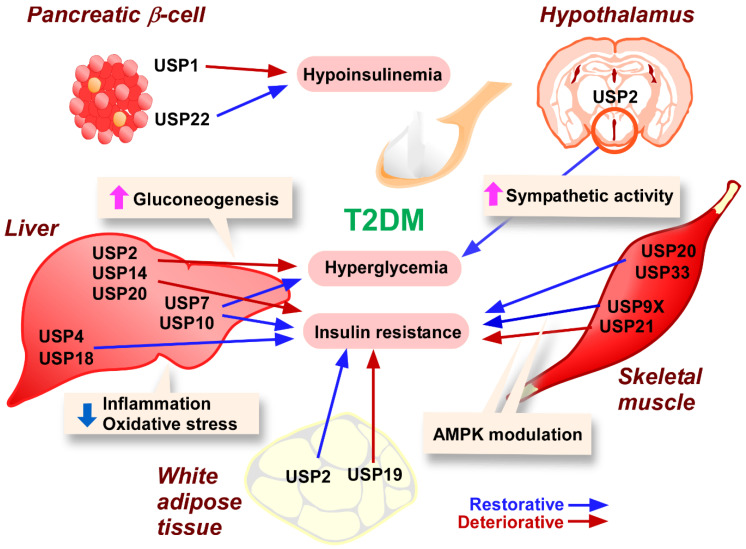
Roles of ubiquitin-specific proteases (USPs) in type 2 diabetes mellitus (T2DM). Blue and red lines represent restorative and deteriorative effects on T2DM, respectively.

**Figure 2 ijms-24-03219-f002:**
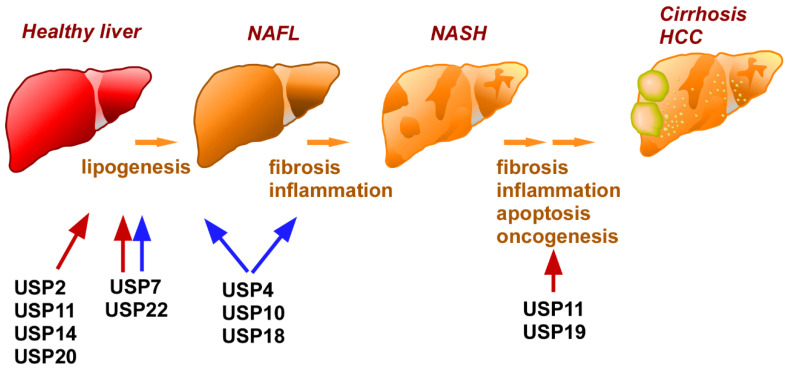
Roles of USPs in non-alcoholic fatty liver disease (NAFLD). NAFL, NASH, and HCC stand for non-alcoholic fatty liver, non-alcoholic steatohepatitis, and hepatocellular carcinoma, respectively. Blue and red arrows represent the restorative and deteriorative effects, respectively, of various USPs on T2DM.

**Figure 3 ijms-24-03219-f003:**
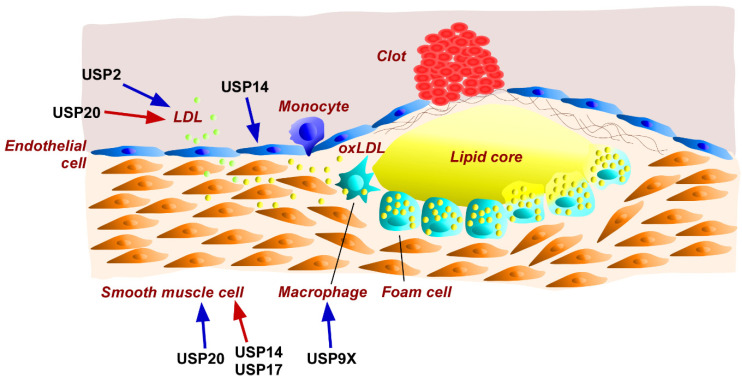
The roles of USPs in atherosclerosis. Blue and red arrows represent restorative and deteriorative effects, respectively, on atherosclerosis. oxLDL stands for oxidative LDL.

**Table 1 ijms-24-03219-t001:** Pathophysiological roles of USPs in metabolic disorders.

Disease	USPs	Expressing Cells/Tissues	Restorative/Deteriorative	Direct Targets	References
Obesity	USP2	Adipocyte	Deteriorative	Cyclin D	[46]
	USP7	Adipocyte	Deteriorative	SREBP1c	[45]
	USP15	Adipocyte	Deteriorative	Keap1	[42]
	USP19	Adipocyte	Deteriorative	?	[49]
	USP20	Hepatocyte	Deteriorative	HMGCR	[50]
	USP53	Adipose tissue	Restorative	?	[53]
T1DM	USP18	β-cell	Restorative	?	[58]
	USP22	Treg	Restorative	FoxP3	[56]
T2DM	USP1	β-cell	Deteriorative	?	[69]
	USP22	β-cell	Restorative	Sirt-1	[68]
	USP2	Adipose tissue macrophage	Restorative	?	[78,79]
	USP19	Adipocyte	Deteriorative	?	[49]
	USP9X	Myocyte	Restorative	AMPKα2	[87]
	USP20	Myocyte	Restorative	β_2_ adrenoreceptor	[94]
	USP21	Myocyte	Deteriorative	DNA-PKcs, ACLY	[84]
	USP33	Myocyte	Restorative	β_2_ adrenoreceptor	[94,95]
	USP2	Hepatocyte	Deteriorative	C/EBPα	[102]
	USP4	Hepatocyte	Restorative	TAK1	[109]
	USP7	Hepatocyte	Restorative	IRS1	[99]
			Restorative	PPARγ	[100]
			Restorative	FoxO1	[101]
	USP10	Hepatocyte	Restorative	Sirt-6	[114]
	USP14	Hepatocyte	Deteriorative	CBP	[103]
			Deteriorative	FASN	[108]
	USP18	Hepatocyte	Restorative	TAK1	[111]
	USP20	Hepatocyte	Deteriorative	HMGCR	[50]
	USP2	Hypothalamus	Restorative	?	[117]
Diabetic nephropathy	USP2	Kidney	?	?	[130]
	USP7	Kidney	?	H2A, H2B?	[132]
	USP9X	Renal epithelial cell	Restorative	Connexin43	[139]
		Mesangial cell	Restorative	Nrf2	[141]
	USP14	Podocyte	Deteriorative	SPAG5-AS	[147]
	USP15	Podocyte	Deteriorative	Keap1	[150]
	USP16	Kidney	?	H2A, H2B?	[132]
	USP21	Kidney	?	H2A, H2B?	[132]
	USP22	Kidney	?	H2A? H2B	[132]
		Mesangial cell	Restorative	Sirt-1	[135]
		Renal epithelial cell	Restorative	Sirt-1	[136]
		Kidney	Deteriorative	?	[137]
		Podocyte	Deteriorative	?	[138]
	USP36	Renal epithelial cell	Deteriorative	DOCK4	[144]
Diabetic retinopathy	USP1	Vascular endothelial cells	Deteriorative	?	[160]
	USP14	Müller cells	Deteriorative	TGF-β receptor, IκBα, Nrf2	[162]
	USP48	Pigment epithelial cell	Restorative	NFκBp65	[159]
Diabetic neuropathy	USP5	Spinal dorsal horn	Deteriorative	Cav3.2	[167]
Diabetic myopathy	USP19	Myocyte	?	?	[174]
	USP21	Myocyte	Deteriorative	DNA-PKcs, ACLY	[84]
Diabetic cardiomyopathy	USP10	Cardiomyocyte	Restorative	NICD1	[176]
Diabetic foot ulcers	USP7	Vascular endothelial cells	Deteriorative	p53	[179]
NAFLD	USP2	Hepatocytes	Deteriorative	FASN	[198,199]
	USP4	Hepatocytes	Restorative	TAK1	[109]
	USP7	Hepatocyte	Restorative	IRS1	[99]
			Deteriorative	Mdm2	[185]
			Deteriorative	PPARγ	[100]
			Deteriorative	ZNF638,	[184]
				CREB,	
				SREBP1c	
	USP10	Hepatocyte	Restorative	Sirt-6	[114]
			Restorative	?	[191]
	USP11	Hepatocyte	Deteriorative	KLF4	[201]
	USP14	Hepatocyte	Deteriorative	FASN	[108]
	USP18	Hepatocyte	Restorative	TAK1	[111]
	USP19	Hepatocyte	Deteriorative	SOAT1	[204]
	USP20	Hepatocyte	Deteriorative	HMGCR	[50]
	USP22	Hepatocyte	Restorative	Sirt-1	[195]
		Hepatocyte	Deteriorative	PPARγ	[193]
Atherosclerosis	USP2	Hepatocyte	Restorative	IDOL	[212]
	USP9X	Macrophage	Restorative	SR-A1	[223]
	USP14	Vascular endothelial cell	Restorative	NLRC5	[221]
		Vascular smooth muscle cell	Deteriorative	PMSD7	[217]
	USP17	Vascular smooth muscle cell	Deteriorative	?	[224]
	USP20	Vascular smooth muscle cell	Restorative	TRAF6	[215]
			Restorative	RIPK1	[216]
		Liver	Deteriorative?	HMGCR	[50]
Cushing disease	USP8	Corticotroph adenoma	Deteriorative	?	[228,229,230,231,232,233,234,235,236,237,238]
	USP48	Corticotroph adenoma	Deteriorative	?	[237]

Metabolic disorders, USP species, USP-expressing cells or tissues, pathological roles of USPs, direct molecular targets, and references are shown.

## Data Availability

Not applicable.

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
