# Peer review of "Ubiquitin-Specific Proteases (USPs) and Metabolic Disorders"

_ijms, 2023, doi:10.3390/ijms24043219_

Round 1
Reviewer 1 Report
Metabolic disorders present a substantial challenge to the health systems of industrialized countries and research into their genesis and prevention is of highest importance. In his review, Prof. Kitamura has summarized the current knowledge about the effect of one specific class of deubiquitinases, USPs, on metabolic disorders including among others obesity, diabetes, steatohepatitis, and atherosclerosis. The breadth of the topic presents a major challenge, as biochemical, cell biological, in vivo experimental as well as medical data have to be summarized in order to give a useful overview of the state of research. The task is further complicated by the incompleteness of the available information. I find the review well-written and successful at its task. The illustrations are clear and the table is very helpful and a great asset for the interested reader.
I highly appreciate that the author highlights the complexity (e.g. USP2 expression in different tissues) of the systems investigated. A fundamental problem, also thankfully highlighted by the author, is that USP inhibitors for a specific USP are often not well characterized with regards to specificity on other USPs and on off-site inhibition or interaction with other, non-USP proteins. As a result, it is rarely possible to present a complete chain of causation ranging from activity of a particular USP to changes in cellular signaling to alterations of metabolic pathways to the medical symptoms. The author frequently mentions whether up- or downregulation of a certain USP exacerbates or remedies a medical symptom, but sometimes this information is omitted and it is simply stated that 'USP20 promotes obesity'. I suspect that upregulation or increased activity are usually referred to when a USP is described as exacerbating or rectifying, but it would be really helpful if it was explicitly stated whether the abundance or activity of the USP in question changed.
Two paragraphs of the review propose that traditional chinese medicine herbal preparations act on specific USPs, with limited evidence. Gastrodin, a molecule found in orchid extract, is described as a panacea against western diet-induced metabolic issues in mice, based on one paper. The connection to USP4 is introduced with an exceptional degree of caution by the author of the review, although the connection is made quite explicitly by the authors of the original paper. Since Gastrodin is a defined chemical molecule that can be purchased in pure form, biochemical assay data on USP4 inhibition should be available in the literature if this connection is really suspected, but I haven't found it either. I would suggest to mention that at present there is only a single study making this connection, without biochemical assay evidence.
In another section, ethanol extracts of artemisia dracunculus are discussed in context of diabetic myopathy and modulated expression of two USPs in two specific muscles are mentioned. These extracts are very likely a mixture of thousands of organic molecules and it seems unlikely that they affect primarily two USPs in two nerves of the entire organism. The original paper (PMID: 24985101) shows that only those two muscles were examined (as opposed to all the other mouse tissues) and that a multitude of changes in protein expression were discovered ('Although MuRF-1 ubiquitin ligase gene expression is consistently down-regulated in skeletal muscle, atrogin-1, Fbxo40, and Traf6 expression is differentially regulated by PMI 5011. Genes encoding other enzymes of the ubiquitin-proteasome system ranging from ubiquitin to ubiquitin-specific proteases are also regulated by PMI 5011.'). My concern is that the way this study is presented in the review suggests a degree of precision that this treatment method is incapable of achieving. The type of study where a highly complex mixture of compounds (plant extract) is thrown at a highly complex system (rodent) can serve as the initial point of investigation, but until chemically defined active ingredients are verified and biochemically characterized, no statements on inhibition of specific proteins can be made.
In summary, if these very minor issues can be addressed, I fully support publication of this review as it presents an excellent overview of a very broad and complex topic of eminent medical importance.
Author Response
Responses to the comments from Reviewer #1
I greatly appreciate the reviewer’s insightful comments, and have revised the manuscript accordingly. Revisions are indicated by red text in the manuscript.
Comment #1: “The author frequently mentions whether up- or downregulation of a certain USP exacerbates or remedies a medical symptom, but sometimes this information is omitted and it is simply stated that 'USP20 promotes obesity'. I suspect that upregulation or increased activity are usually referred to when a USP is described as exacerbating or rectifying, but it would be really helpful if it was explicitly stated whether the abundance or activity of the USP in question changed.”
My Response: I completely agree with this reviewer’s suggestion. In the revised text, I clearly state whether the abundance or activity of the USPs is altered in the cited reports as follows:
- “The authors also suggested that USP15 promotes adipogenesis because the overexpression of USP15 protects a CRL component, Keap1, from autoubiquitination [42,43].” (Lines90-92)
- “Because CRISPR/Cas9-dependent KO of the Usp22 gene decreased the occurrence of fork-head box P3 (FoxP3) + cells, it appears that USP22, a deubiquitinating enzyme in the SAGA chromatin modifying complex, stabilizes FoxP3 [56].” (Lines 153-155)
- “Santin et al. reported that siRNA against USP18 increased type I IFN–elicited events in beta-cells, such as signal transducer and activator of transcription (STAT) 1/2–induced inflammatory responses, and mitochondrial pathway-driven apoptosis [58]. Moreover, Usp18 knockdown also strengthened chemokine induction via the increase of melanoma differentiation-associated protein 5 (MDA5), which is a sensor protein of viral double-stranded RNA [58].”(Lines 160-165)
- “Kitamura et al. reported that USP2 knockdown in human macrophage-like HL-60 cells displayed increased expression of proinflammatory cytokines in response to inflammatory stimuli, while macrophages isolated from Usp2 transgenic mice had repressed cytokine expression [77].” (Lines 235-238)
- “Overexpression of Usp21 interfered with proteasome-dependent digestion of DNA-PKcs and ACLY via removal of the polyubiquitin chain [84]. Subsequently, accumulated DNA-PK and ACLY blunted the activation of AMP-activated kinase (AMPK), which is responsible for mitochondrial fuel activation [84].” (Lines 265-268)
- “In this case, siRNA for Usp9X, but not Usp5 and 7, blocked ubiquitination-mediated degradation of the AMPKa2 subunit [87].” (Lines 275-277)
- “A previous paper employing overexpression and knockdown models demonstrated that USP20 and 33 have the potential to remove polyubiquitin chains from the b2 adrenoreceptor [94].” (Lines 283-285)
- “Moreover, overexpression of Usp7decreased polyubiquitination of FoxO1, a transcription factor that regulates the transcription of the rate-limiting enzymes for gluconeogenesis, including glucose 6-phosphatase catalytic subunit (G6PC) and phosphoenolpyruvate carboxykinase 1 (PCK1) [101].” (Lines 315-318)
- “At the mechanistic level, overexpression of Usp2 was shown to stabilize C/EBP-alpha, a transcription factor that induces the expression of 11beta-hydroxysteroid dehydrogenase 1 (HSD1) [102].“ (Lines 332-334)
- “Overexpression and knockdown of Usp14 respectively increased and decreased 3’,5’-cyclic monophosphate-responsive element binding protein (CBP) in the liver of obese mice [103].” (Lines 344-346)
- “Assessments using an adenoviral Usp18 and hepatocyte-selective Usp18KO mice revealed that USP18 suppressed the TAK1-NF-kappaB/JNK signaling axis [111]. “(Lines 370-372)
- “As mentioned in Section 2, obesity induced by a 23-week high-fat and high-sucrose diet was mitigated in liver-specific Usp20KO mice, indicating liver-specific Usp20 deficiency caused significant restoration of glucose and insulin tolerance [50].” (Lines 396-399)
- “Mechanistically, a ML364 treatment experiment suggests that USP2 inhibits the accumulation of ROS in VMH neurons, and thereby leads to continued ATP production in the mitochondria.” (Lines 410-413)
- “Overexpression of Usp9x in these cells attenuated glucose-induced epithelial-to-mesenchymal transition (EMT) in a DUB activity–dependent manner [140], which is a pathology associated with diabetic nephropathy [139].” (Lines 494-496)
- “Overexpression of Usp9x has been shown to stabilize Nrf2 and to increase Hox1 and Nqo1 transcription [141].” (Lines 506-507)
- “In addition, increased expression of Usp14 enhances inflammatory cytokine production in Müller cells in diabetic conditions via the activation of NF-kB signaling through increased IkappaBalpha degradation [162]. Moreover, introduction of a Usp14-expressing construct increased ROS generation in Müller cells through marked suppression of superoxide dismutase activity and the repression of several antioxidant proteins [162]. (Lines 577-582)
- “Lee et al. reported that overexpression of USP7 directly stabilizes PPARgamma, a pivotal adipogenic transcription factor, in COS7 cells [100]. Additionally, USP7 overexpression upregulated PPARG mRNA in HepG2 hepatocytes, which was followed by increases in glucose and fatty acid uptake [100]. Moreover, infection of Usp7-expressing adenovirus increased PPARg levels in the liver of mice, and accelerated the development of fatty liver [100].” (Lines 685-690).
- “By employing overexpression and knockdown Usp7 models, they demonstrated that USP7 increased ZNF638 protein expression in SK-Hep1 hepatocytes by two distinct mechanisms: direct stabilization by deubiquitination of ZNF638, and induction of ZNF638 mRNA by stabilization of cAMP-responsive element binding protein (CREB), an upstream regulator [184].” (Lines 693-697)
- “Because treatment with siRNA for USP22 diminished G protein alpha12–induced accumulation of Sirt-1 in HepG2 cells, it was concluded that UPS22 stabilizes Sirt-1 [195].” (Lines 761-763)
- “Experiments using models of USP19 overexpression and knockdown demonstrated that USP19 decreased K33/K48-linked ubiquitination at the 120th lysine in SORT1 and prevented its digestion by the proteasome in hepatoma cell lines [205].” (Lines 814-817)
- “Since overexpression of Usp2 promotes the deubiquitination of IDOL, USP2 was considered to suppress LDL uptake via ubiquitin-dependent degradation of LDL [212].” (Lines 840-842)
- “Transfection of a Usp2-expressing construct restored IDOL-repressed LDL uptake in cultured cells, whereas Usp2 knockdown reduced LDL uptake by ~50% in HepG2 hepatocytes [212].” (Lines 844-846)
- “Overexpression and gene interference experiments indicated that USP9X digests the K63 polyubiquitin chain at the K27 residue of class A1 scavenger receptor (SR-A1), but not other scavenger receptors in cultured cells [223].” (Lines 907-909)
Comment#2: “Two paragraphs of the review propose that traditional chinese medicine herbal preparations act on specific USPs, with limited evidence. Gastrodin, a molecule found in orchid extract, is described as a panacea against western diet-induced metabolic issues in mice, based on one paper. The connection to USP4 is introduced with an exceptional degree of caution by the author of the review, although the connection is made quite explicitly by the authors of the original paper. Since Gastrodin is a defined chemical molecule that can be purchased in pure form, biochemical assay data on USP4 inhibition should be available in the literature if this connection is really suspected, but I haven't found it either. I would suggest to mention that at present there is only a single study making this connection, without biochemical assay evidence.
In another section, ethanol extracts of artemisia dracunculus are discussed in context of diabetic myopathy and modulated expression of two USPs in two specific muscles are mentioned. These extracts are very likely a mixture of thousands of organic molecules and it seems unlikely that they affect primarily two USPs in two nerves of the entire organism. The original paper (PMID: 24985101) shows that only those two muscles were examined (as opposed to all the other mouse tissues) and that a multitude of changes in protein expression were discovered ('Although MuRF-1 ubiquitin ligase gene expression is consistently down-regulated in skeletal muscle, atrogin-1, Fbxo40, and Traf6 expression is differentially regulated by PMI 5011. Genes encoding other enzymes of the ubiquitin-proteasome system ranging from ubiquitin to ubiquitin-specific proteases are also regulated by PMI 5011.'). My concern is that the way this study is presented in the review suggests a degree of precision that this treatment method is incapable of achieving. The type of study where a highly complex mixture of compounds (plant extract) is thrown at a highly complex system (rodent) can serve as the initial point of investigation, but until chemically defined active ingredients are verified and biochemically characterized, no statements on inhibition of specific proteins can be made. “
My Response: As the reviewer pointed out, two reports did not provide sufficient data for concluding that gastrodin or pure chemicals in Artemisia dracunculus selectively affect molecules that determine the progression of metabolic disorders. Therefore, I have revised the manuscript as follows:
- I have focused on the effects of gastrodin on USP4, and have removed mention of other information including the protective effects of pancreatic beta-cells. Additionally, I describe and elaborate on experimental results concerning the effects of gastrodin on USP4.
“A report suggested that USP4 might also mediate the beneficial effects of traditional medicine in the context of insulin signaling. Gastrodin, a chemical derived from Gastrodia elata Blume (Orchidaceae), significantly decreased dexamethasone-induced ubiquitination of insulin receptor in HepG2 cells, and concomitant increases of USP4 mRNA and protein were also observed [113]. Mechanistically, gastrodin suppressed the association of GATA binding protein 1 (GATA1) with the Usp4 promoter, suggesting gastrodin likely increases USP4 transcription in a GATA1-dependent manner [113]. Thus, gastrodin might avoid the proteasome-dependent digestion of insulin receptor in hepatocytes via USP4 induction.“(Lines 378-386)
- As the reviewer suggested, these findings do not exclude the possibility that changes in expression of USP14 and USP19 are not correlated with the effect of ethanol extract of Artemisia dracunculus. Thus, I have deleted this section from the revised manuscript.
English language and style
The original submitted manuscript and the revised manuscript were both edited by two native English speakers working for an academic English editing service.

Reviewer 2 Report
The MS is devoted to the description of ubiquitin-specific proteases (USPs) participation in different metabolic disorders. The MS is skillfully written and well illustrated. It may be interesting for many investigators in this field. At the same time, I have a remark.
It should be more clearly noted that the USPs functions are for the most part connected with UPS (Ubiquitin Proteasome System) functions. The MS contains several examples of this (lines 782-785, 822-824). Despite this, I would like to see a clearer piece of information about USPs and UPS interaction. For example, it should be pointed out that USP14 often described by the author, is a part of the 26S proteasome structure.
Author Response
Responses to the comments from Reviewer #2
I appreciate the reviewer’s insightful comments and have revised the manuscript accordingly. All revisions are indicated by red text in the main manuscript.
Comment: “It should be more clearly noted that the USPs functions are for the most part connected with UPS (Ubiquitin Proteasome System) functions. The MS contains several examples of this (lines 782-785, 822-824). Despite this, I would like to see a clearer piece of information about USPs and UPS interaction. For example, it should be pointed out that USP14 often described by the author, is a part of the 26S proteasome structure.”
My Response: To my knowledge, there is little information about regulatory roles of USPs on UPS in metabolic disease. In the manuscript, I discussed the roles of USP14 in one paragraph in the Perspectives section. In this paragraph, I mention that modulation of the proteasome alters the activation of mTOR pathway, which is a determinant of metabolic disorders.
“Although many papers have demonstrated that the regulatory roles of USPs via the stabilization of target proteins determine the incidence of metabolic diseases, the dynamics of the interaction between USPs and the proteasome in metabolic diseases are not fully understood. USP14 is the only USP protein that reversibly associates with 19S regulatory particle of the proteasome and promotes its maturation [242–244], suggesting that USP14 intrinsically controls ubiquitination of target protein as well as proteasomal activity [245]. Full-length USP14 or the UBL domain of USP14 blocks target digestion by the proteasome via an allosteric mechanism [245], whereas the ligation of an ubiquitylated target to USP14 potentiates proteasomal ATPase activity and 20S core particle opening, followed by proteasomal digestion [245]. With respect to the modulation of key molecules for metabolic diseases, such as FASN and CBP [103,221], USP14 might not only disassemble the ubiquitin chain on target proteins, but might also affect the incorporation of the targets into the proteasome. Interestingly, USP14 binds PSMD7, a regulatory subunit of 19S proteasome [217]. Knockdown of PSMD7 dramatically decreased proteasome activity and repressed mTOR/p70S6 kinase [218]. Further, Usp14 knockdown attenuated phosphorylation of both mTOR and p70S6 kinase after PDGF-BB stimulation [217]. mTOR affects cellular metabolism by various pathways including protein synthesis, lipid synthesis, and energy metabolism [246]. Given that changes in mTOR activity also influence incidence of metabolic disease at an individual level [246], regulatory links between the proteasome and mTOR might be therapeutic targets for metabolic disorders. In this respect, further studies investigating the roles of USPs on UPS are required.” (Lines 1043-1063)
English language and style
The original submitted manuscript and the revised manuscript were both edited by two native English speakers working for an academic English editing service.
